# Chimera: Effectively Modeling Multivariate Time Series with 2-Dimensional State Space Models

**Ali Behrouz**
Cornell University
ab2947@cornell.edu

**Michele Santacatterina**
New York University
santam13@nyu.edu

**Ramin Zabih**
Cornell University
rdz@cs.cornell.edu

## Abstract

Modeling multivariate time series is a well-established problem with a wide range of applications from healthcare to financial markets. It, however, is challenging as it requires methods to (1) have high expressive power of representing complicated dependencies along the time axis to capture both long-term progression and seasonal patterns, (2) capture the inter-variate dependencies when it is informative, (3) dynamically model the dependencies of variate and time dimensions, and (4) have efficient training and inference for very long sequences. Traditional State Space Models (SSMs) are classical approaches for univariate time series modeling due to their simplicity and expressive power to represent *linear* dependencies. They, however, have fundamentally limited expressive power to capture non-linear dependencies, are slow in practice, and fail to model the inter-variate information flow. Despite recent attempts to improve the expressive power of SSMs by using deep structured SSMs, the existing methods are either limited to univariate time series, fail to model complex patterns (e.g., seasonal patterns), fail to dynamically model the dependencies of variate and time dimensions, and/or are input-independent. We present Chimera, an expressive variation of the 2-dimensional SSMs with careful design of parameters to maintain high expressive power while keeping the training complexity linear. Using two SSM heads with different discretization processes and input-dependent parameters, Chimera is provably able to learn long-term progression, seasonal patterns, and desirable dynamic autoregressive processes. To improve the efficiency of complex 2D recurrence, we present a fast training using a new 2-dimensional parallel selective scan. Our experimental evaluation shows the superior performance of Chimera on extensive and diverse benchmarks, including ECG and speech time series classification, long-term and short-term time series forecasting, and time series anomaly detection.

## 1 Introduction

Modeling time series is a well-established problem with a wide range of applications from healthcare [1–3] to financial markets [4, 5] and energy management [6]. The complex nature of time series data, its diverse domains of applicability, and its broad range of tasks (e.g., classification [2, 7], imputation [8, 9], anomaly detection [2, 10], and forecasting [6]), however, raise fundamental challenges to design effective and generalizable models: (1) The higher-order, seasonal, and long-term patterns in time series require an effective model to be able to expressively capture complex and autoregressive dependencies; (2) In the presence of multiple variates of time series, an effective model need to capture the complex dynamics of the dependencies between time and variate axes. More specifically, most existing multivariate models seem to suffer from overfitting especially when the target time series is not correlated with other covariates [11]. Accordingly, an effective model needs to adaptively learn to select (resp. filter) informative (resp. irrelevant) variates; (3) The diverse set of domains and tasks requires effective models to be free of manual pre-processing and domain

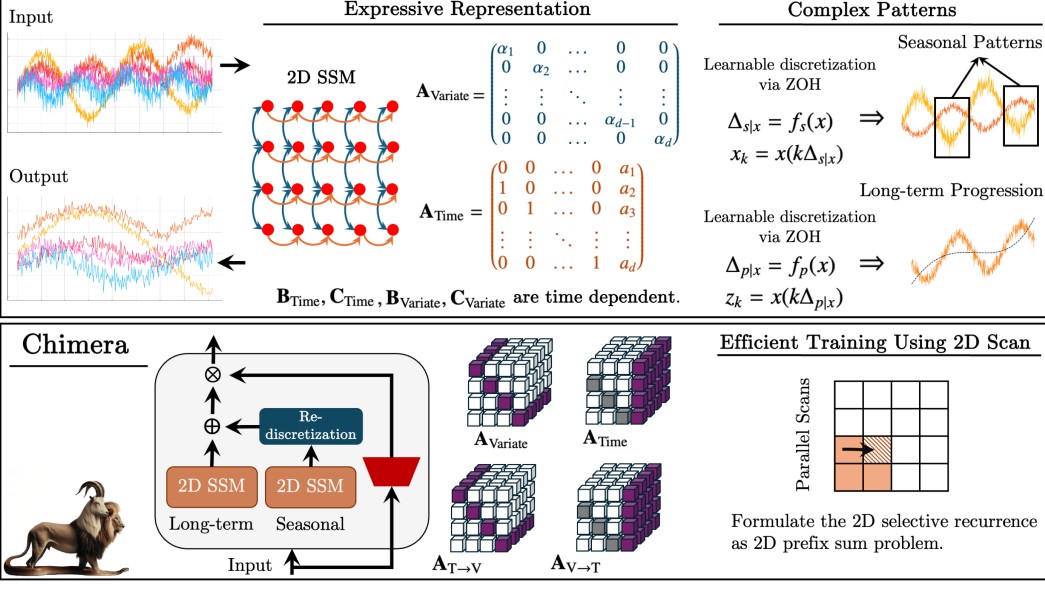

Figure 1: **The Overview of Contributions and Architecture of Chimera.** We present a 2-dimensional SSM with careful and expressive parameterization. It uses different learnable discretization processes to learn seasonal and long-term progression patterns, and leverages a parallelizable and fast training process by re-formulating the 2D input dependent recurrence as a 2D prefix sum problem.

knowledge and instead adaptively learn them; and (4) Due to the processing of very long sequences, effective methods need efficient training and inference.

Classical methods (e.g., State Space Models [12, 13], ARIMA [14], SARIMA [15], Exponential Smoothing (ETS) [16]) require manual data preprocessing and model selection, and often are not able to capture complex *non-linear* dynamics. The raise of deep learning methods and more specifically Transformers [17] has led to significant research efforts to address the limitation of classical methods and develop effective deep models [18–27]. Unfortunately, most existing deep models struggle to achieve all the above four criteria. The main body of research in this direction has focused on designing attention modules that use the special traits of time series [21, 20]. However, the inherent permutation equivariance of attentions contradicts the causal nature of time series and often results in suboptimal performance compared to simple linear methods [11]. Moreover, they often either overlook difference of seasonal and long-term trend or use non-learnable methods to handle them [20].

A considerable subset of deep models overlook the importance of modeling the dependencies of variates [11, 28, 29]. These dependencies, however, are not always useful; specifically when the target time series is not correlated with other covariates [30]. Despite several studies exploring the importance of learning cross variate dependencies [26, 27, 30], there has been no universal standard and the conclusion has been different depending on the domain and benchmarks. Accordingly, we argue that an effective model need to *adaptively* learn to capture the dependencies of variates in a data-dependent manner. In this direction, recently, Liu et al. [27] argue that attention mechanisms are more effective when they are used across variates, showing the importance of modeling complex non-linear dependencies across the variate axis in a data-dependent manner. However, the quadratic complexity of Transformers challenges the model on multivariate time series with a large number of variates (e.g., brain activity signals [2] or traffic forecasting [6]), limiting the efficient training and inference (see Table 3, and Table 5).

The objective of this study is to develop a provably expressive model for multivariate time series that not only can model the dynamics of the dependencies along both time and variates, but it also takes advantage of fast training and inference. To this end, we present a Chimera, a three-headed two-dimensional State Space Model (SSM) that is based on linear layers along (i) time, (ii) variates, (iii) time→variate, and (iv) variate→time. Chimera has a careful parameterization based on the pair of companion and diagonal matrices (see Figure 1), which is provably expressive to recover both classical methods [16, 14, 15], linear attentions, and recent SSM-based models [31, 32]. It further uses an adaptive module based on a 2D SSM with an especially designed discretization process to

capture seasonal patterns. While our theoretical results and design of Chimera guarantee the first three criteria of an effective model, due to its 2D recurrence, the naive implementation of Chimera results in slow training. To address this issue, we reformulate its 2D recurrence as the prefix sum problem with a 2-dimensional associative operators. This new formulation can be done in parallel and has hardware-friendly implementation, resulting in much faster training and inference.

In our experimental evaluation, we explore the performance of Chimera in a wide range of tasks: ECG and audio speech time series classification, long- and short-term time series forecasting, and anomaly detection tasks. We find that Chimera achieve superior or on par performance with state-of-the-art methods, while having faster training and less memory consumption. We perform a case study on the human brain activity signals [2] to show (1) the effectiveness of Chimera and (2) evaluate the importance of modeling the dynamics of the variates dependencies.

## 2 Preliminaries

**Notations.** In this paper we mainly focus on classification and forecasting tasks. Note that anomaly detection can be seen as a binary classification task, where 0 means "normall" and 1 means "anomaly". We let $\mathbf{X} = \{\mathbf{x}_1, \ldots, \mathbf{x}_N\} \in \mathbb{R}^{N \times T}$ be the input sequences, where $N$ is the number of variates and $T$ is the time steps. We use $\mathbf{x}_{v,t}$ to refer to the value of the series $v$ at time $t$. In classification (anomaly detection) tasks, we aim to classify input sequences and for forecasting tasks, given an input sequence $\mathbf{x}_i$, we aim to predict $\tilde{\mathbf{x}}_i \in \mathbb{R}^{1 \times H}$, i.e., the next $H$ time steps for variate $\mathbf{x}_i$, where $H$ is called horizon. In 2D SSMs formulation, for a 2-dimensional vector $x \in \mathbb{C}^1$, we use $x^{(1)}$ and $x^{(2)}$ to refer to its real and imaginary components, respectively.

**Multi-Dimensional State Space Models.** We build our approach on the continuous State Space Model (SSM) but later we make each component of Chimera discrete by a designed discretization process. For additional discussion on 1D SSMs see Appendix A. Given parameters $\mathbf{A}_{\tau_1} \in \mathbb{R}^{N^{(\tau_1)} \times N^{(\tau_1)}}$, $\mathbf{B}_{\tau_2} \in \mathbb{C}^{N^{(\tau_2)} \times 1}$, and $\mathbf{C} \in \mathbb{C}^{N_1 \times N_2}$ for $\tau_1 \in \{1, ..., 4\}$ and $\tau_2 \in \{1, 2\}$, the general form of the time-invariant 2D SSM is the map $\mathbf{x} \in \mathbb{C}^1 \mapsto \mathbf{y} \in \mathbb{C}^1$ defined by the linear Partial Differential Equation (PDE) with initial condition $h(0, 0) = 0$:

$$\frac{\partial}{\partial t^{(1)}} h\left(t^{(1)}, t^{(2)}\right) = \left(\mathbf{A}_1 h^{(1)}\left(t^{(1)}, t^{(2)}\right), \mathbf{A}_2 h^{(2)}\left(t^{(1)}, t^{(2)}\right)\right) + \mathbf{B}_1 \mathbf{x}\left(t^{(1)}, t^{(2)}\right), \quad (1)$$

$$\frac{\partial}{\partial t^{(2)}} h\left(t^{(1)}, t^{(2)}\right) = \left(\mathbf{A}_3 h^{(1)}\left(t^{(1)}, t^{(2)}\right), \mathbf{A}_4 h^{(2)}\left(t^{(1)}, t^{(2)}\right)\right) + \mathbf{B}_2 \mathbf{x}\left(t^{(1)}, t^{(2)}\right), \quad (2)$$

$$\mathbf{y}\left(t^{(1)}, t^{(2)}\right) = \langle \mathbf{C}, \mathbf{x}\left(t^{(1)}, t^{(2)}\right) \rangle. \quad (3)$$

Contrary to the multi-dimensional SSMs discussed by Gu and Dao [33], Gu et al. [34], in which multi-dimension refers to the dimension of the input but with one time variable, the above formulation uses two variables, meaning that the mapping is from a 2D grid to a 2D grid.

**(Seasonal) Autoregressive Process.** Autoregressive process is a basic yet essential premise for time series modeling, which models the causal nature of time series. Given $p \in \mathbb{N}$, $\mathbf{x}_k \in \mathbb{R}^d$, the simple linear autoregressive relationships between $\mathbf{x}_k$ and its past samples $\mathbf{x}_{k-1}, \mathbf{x}_{k-2}, \ldots, \mathbf{x}_{k-p}$ can be modeled as $\mathbf{x}_k = \phi_1 \mathbf{x}_{k-1} + \phi_2 \mathbf{x}_{k-2} + \ldots, \phi_p \mathbf{x}_{k-p}$, where $\phi_1, \ldots, \phi_p$ are coefficients. This is called AR($p$). Similarly, in the presence of seasonal patterns, the seasonal autoregressive process, SAR($p, q, s$), is:

$$\mathbf{x}_k = \phi_1 \mathbf{x}_{k-1} + \phi_2 \mathbf{x}_{k-2} + \ldots, \phi_p \mathbf{x}_{k-p} + \eta_1 \mathbf{x}_{k-s} + \eta_2 \mathbf{x}_{k-2s} + \cdots + \eta_q \mathbf{x}_{k-qs}, \quad (4)$$

where $s$ is the frequency of seasonality, and $\phi_1, \ldots, \phi_p$ and $\eta_1, \ldots, \eta_q$ are coefficients. Note that one can simply extend the above formulation to multivariate time series by letting coefficients to be vectors and replace the product with element-wise product.

## 3 Chimera: A Three-headed 2-Dimensional State Space Model

In this section, we first present a mathematical model for multivariate time series data and then based on this model, we present a neural architecture that can satisfy all the criteria discussed in §1.

## 3.1 Motivations & Chimera Model

SSMs have been long-standing methods for modeling time series [12, 13], mainly due to their simplicity and expressive power to represent complicated and autoregressive dependencies. Their states, however, are the function of a single-variable (e.g., time). Multivariate time series, on the other hand, require capturing dependencies along both time and variate dimensions, requiring the current state of the model to be the function of both time and variate. Classical 2D SSMs [35–38], however, struggle to achieve good performance compared to recent advanced deep learning methods as they are : (1) only able to capture linear dependencies, (2) discrete by design, having a pre-determined resolution, and so cannot simply model seasonal patterns, (3) slow in practice for large datasets, (4) their update parameters are static and cannot capture the dynamics of dependencies. Deep learning-based methods [6, 30, 27], on the other hand, potentially are able to address a subset of the above limitations, while having their own drawbacks (discussed in §1). In this section, we start with *continuous* SSMs due to their connection to both classical methods [12, 13] and recent breakthrough in deep learning [34, 33]. We then discuss our contributions on how to take the advantages of the best of both worlds, addressing all the abovementioned limitations.

**Discrete 2D SSM.** We use 2-dimensional SSMs, introduced in Equation 1-3, to model multivariate time series, where the first axis corresponds to the time dimension and the second axis is the variates. Accordingly, each state is a function of both time and variates. The first stage is to transform the continuous form of 2D SSMs to discrete form. Given the step size $\Delta_1$ and $\Delta_2$, which represent the resolution of the input along the axes, discrete form of the input is defined as $\mathbf{x}_{k,\ell} = \mathbf{x}(k\Delta_1, \ell\Delta_2)$. Using Zero-Order Hold (ZOH) method, we can discretize the input as (see Appendix C for details):

$$\begin{pmatrix} h_{k,\ell+1}^{(1)} \\ h_{k+1,\ell}^{(2)} \end{pmatrix} = \begin{pmatrix} \bar{\mathbf{A}}_1 & \bar{\mathbf{A}}_2 \\ \bar{\mathbf{A}}_3 & \bar{\mathbf{A}}_4 \end{pmatrix} \begin{pmatrix} h_{k,\ell}^{(1)} \\ h_{k,\ell}^{(2)} \end{pmatrix} + \begin{pmatrix} \bar{\mathbf{B}}_1 \\ \bar{\mathbf{B}}_2 \end{pmatrix} \otimes \begin{pmatrix} \bar{\mathbf{x}}_{k,\ell+1} \\ \bar{\mathbf{x}}_{k+1,\ell} \end{pmatrix}, \tag{5}$$

where $\bar{\mathbf{A}}_i = \exp\left(\Delta_{\lfloor \frac{i+1}{2} \rfloor} \mathbf{A}_i\right)$ for $i = 1, 2, 3, 4$, $\bar{\mathbf{B}}_1 = \begin{pmatrix} \mathbf{A}_1^{-1} \left(\bar{\mathbf{A}}_1 - \mathbf{I}\right) \mathbf{B}_1^{(1)} \\ \mathbf{A}_2^{-1} \left(\bar{\mathbf{A}}_2 - \mathbf{I}\right) \mathbf{B}_1^{(2)} \end{pmatrix}$, and $\bar{\mathbf{B}}_2 = \begin{pmatrix} \mathbf{A}_3^{-1} \left(\bar{\mathbf{A}}_3 - \mathbf{I}\right) \mathbf{B}_2^{(1)} \\ \mathbf{A}_4^{-1} \left(\bar{\mathbf{A}}_4 - \mathbf{I}\right) \mathbf{B}_2^{(2)} \end{pmatrix}$. Note that this formulation can also be viewed as the modification of the discrete Roesser's SSM model [35] when we add a lag of 1 in the inputs $\begin{pmatrix} \bar{\mathbf{x}}_{i,j} \\ \bar{\mathbf{x}}_{i,j} \end{pmatrix}$. This modification, however, misses the discretization step, which is an important step in our model. We later use the discretization step to (1) empower the model to select (resp. filter) relevant (resp. irrelevant) information, (2) adaptively adjust the resolution of the method, capturing seasonal patterns.

From now on, we use $t$ (resp. $v$) to refer to the index along the time (resp. variate) dimension. Therefore, for the sake of simplicity, we reformulate Equation 5 as follows:

$$h_{v,t+1}^{(1)} = \bar{\mathbf{A}}_1 h_{v,t}^{(1)} + \bar{\mathbf{A}}_2 h_{v,t}^{(2)} + \bar{\mathbf{B}}_1 \mathbf{x}_{v,t+1}, \tag{6}$$

$$h_{v+1,t}^{(2)} = \bar{\mathbf{A}}_3 h_{v,t}^{(1)} + \bar{\mathbf{A}}_4 h_{v,t}^{(2)} + \bar{\mathbf{B}}_2 \mathbf{x}_{v+1,t}, \tag{7}$$

$$\mathbf{y}_{v,t} = \mathbf{C}_1 h_{v,t}^{(1)} + \mathbf{C}_2 h_{v,t}^{(2)}, \tag{8}$$

where $\bar{\mathbf{A}}_1, \bar{\mathbf{A}}_2, \bar{\mathbf{A}}_3, \bar{\mathbf{A}}_4 \in \mathbb{R}^{N \times N}$, $\bar{\mathbf{B}}_1, \bar{\mathbf{B}}_2 \in \mathbb{R}^{N \times 1}$, and $\mathbf{C}_1, \mathbf{C}_2 \in \mathbb{R}^{1 \times N}$ are parameters of the model, $h_{v,t}^{(1)}, h_{v,t}^{(2)} \in \mathbb{R}^{N \times d}$ are hidden states, and $\mathbf{x}_{v,t} \in \mathbb{R}^{1 \times d}$ is the input. In this formulation, intuitively, $h_{v,t}^{(1)}$ is the hidden state that carries cross-time information (each state depends on its previous time stamp but within the same variate), where $\bar{\mathbf{A}}_1$ and $\bar{\mathbf{A}}_2$ control the emphasis on past cross-time and cross-variate information, respectively. Similarly, $h_{v,t}^{(2)}$ is the hidden state that carries cross-variate information (each state depends on other variates but with the same time stamp). Later in this section, we discuss to modify the model to bi-directional setting along the variate dimension, to enhance information flow along this non-causal dimension.

**Interpretation of Discretization.** Time series data are often sampled from an underlying continuous process [39, 40]. In these cases, variable $\Delta_1$ in the discretization of the time axis can be interpreted as resolution or the sampling rate from the underlying continuous data. However, discretization along

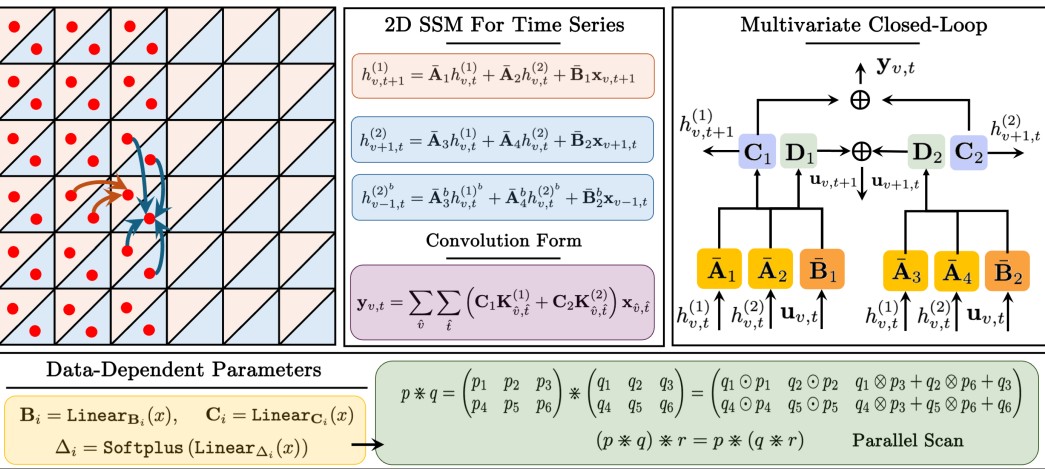

Figure 2: **Different forms of Chimera.** **(Top-Left)** Chimera has a recurrence form (bi-directional along the variates), which also can be computed as a global convolution in training. **(Top-Right)** In forecasting, we present the multivariate closed-loop to improve the performance for long horizons. **(Bottom)** Using data-dependent parameters, Chimera training can be done as a parallel 2D scan.

the variate axis, which is discrete by its nature, or when working directly with discrete data [41] is an unintuitive process, and raise questions about its significance. The discretization step in 1D SSMs has deep connections to gating mechanisms of RNNs [42, 33], automatically ensures that the model is normalized [43], and results in desirable properties such as resolution invariance [32].

**Proposition 3.1.** *The 2D discrete SSM introduced in Equation 6-8 with parameters* $(\{\bar{\mathbf{A}}_i\}, \{\bar{\mathbf{B}}_i\}, \{\bar{\mathbf{C}}_i\}, k\Delta_1, \ell\Delta_2)$ *evolves at a rate $k$ (resp. $\ell$) times as fast as the 2D discrete SSM with parameters* $(\{\bar{\mathbf{A}}_i\}, \{\bar{\mathbf{B}}_i\}, \{\bar{\mathbf{C}}_i\}, \Delta_1, \ell\Delta_2)$ *(resp.* $(\{\bar{\mathbf{A}}_i\}, \{\bar{\mathbf{B}}_i\}, \{\bar{\mathbf{C}}_i\}, k\Delta_1, \Delta_2)$*).*

Accordingly, parameters $\Delta_1$ can be viewed as the controller of the length of dependencies that the model captures. That is, based on the above result, we see the discretization along the time axis as the setting of the resolution or sampling rate: while small $\Delta_1$ can capture long-term progression, larger $\Delta_1$ captures seasonal patterns. For now, we see the discretization along the variate axis as a mechanism similar to gating in RNNs [44, 33], where $\Delta_2$ controls the length of the model context. Larger values of $\Delta_2$ means less context window, ignoring other variates, while smaller values of $\Delta_2$ means more emphasis on the dependencies of variates. Later, inspired by Gu and Dao [33], we discuss making $\Delta_2$ as the function of the input, resulting in a selection mechanism that filters irrelevant variates.

**Structure of Transition Matrices.** For Chimera to be expressive and able to recover autoregressive process, hidden states $h_{v,t}^{(1)}$ should carry information about *past* time stamps. While making all the parameters in $\mathbf{A}_i$ learnable allows the model to learn any arbitrary structure for $\mathbf{A}_i$, previous studies show that this is not possible unless the structure of transition matrices are restricted [45, 46]. To this end, inspired by Zhang et al. [28] that argue that companion matrices are effective to capture the dependencies along the time dimension, we restrict $\mathbf{A}_1$ and $\mathbf{A}_2$ matrices to have companion structure (see Appendix D for the details). Not only this formulation is shown to be effective for capturing dependencies along the time dimension [28] (also see Theorem 3.4), but it also can help us to compute the power of $\mathbf{A}_1$ and $\mathbf{A}_2$ faster in the convolutional form, as discussed by Zhang et al. [28]. Also, for $\mathbf{A}_3$ and $\mathbf{A}_4$, we observe that even a simpler structure of diagonal matrices is effective to fuse information along the variate dimension. Not only these simple structured matrices make the training of the model faster, but they also are proven to be effective [45].

**Bi-Directionality.** The causal nature of the 2D SSM result in limited information flow along the variate dimension as variate are not ordered. To overcome this challenge, inspired by the bi-directional 1D SSMs [47, 48, 1], we use two different modules for forward and backward pass along the variate dimension. The details of formulation is in Appendix E.

**Convolution Form.** As discussed by Baron et al. [49], similar to 1D SSMs [34], the *data-independent* formulation of 2D-SSMs can be viewed as a convolution with a kernel $\mathbf{K}$. This formulation not only results in faster training by providing the ability of parallel processing, but it also connect Chimera

with very recent studies of modern convolution-based architecture for time series [9]. Applying the recurrent rules in Equation 6-8, we can write the output as:

$$\mathbf{y}_{v,t} = \sum_{1 \leq \hat{v} \leq v} \sum_{1 \leq \hat{t} \leq t} \left( \mathbf{C}_1 \mathbf{K}_{\hat{v},\hat{t}}^{(1)} + \mathbf{C}_2 \mathbf{K}_{\hat{v},\hat{t}}^{(2)} \right) \mathbf{x}_{\hat{v},\hat{t}}, \tag{9}$$

where kernels $\mathbf{K}_{\hat{v},\hat{t}}^{(\tau)} = \sum_{(z_1,\ldots,z_5) \in \mathbf{P}^{(\tau)}} q_i \, \bar{\mathbf{A}}_1^{p_1} \bar{\mathbf{A}}_2^{p_2} \bar{\mathbf{A}}_3^{p_3} \bar{\mathbf{A}}_4^{p_4} \bar{\mathbf{B}}_{p_5}$, and $\mathbf{P}^{(\tau)}$ is the partitioning of the paths from the starting point to $(\hat{v}, \hat{t})$ for $\tau \in \{1, 2\}$. As discussed by Baron et al. [49], if the power of $\bar{\mathbf{A}}_i$s are given and cached, calculating the partitioning of all paths can be done very efficiently (near-linearly) as it the generalization of pascal triangle. To calculate the power of $\bar{\mathbf{A}}_i$, note that we use diagonal matrices as the structure of $\bar{\mathbf{A}}_3$, and $\bar{\mathbf{A}}_4$, and so computing their powers is very fast. On the other hand, for $\bar{\mathbf{A}}_1$ and $\bar{\mathbf{A}}_2$ with companion structures, we can use sparse matrix multiplication, which results in linear complexity in terms of the sequence length.

**Data-Dependent Parameters.** As discussed earlier, parameters $\bar{\mathbf{A}}_1$ and $\bar{\mathbf{A}}_2$ controls the emphasis on past cross-time and cross-variate information. Similarly, parameters $\Delta_1$ and $\bar{\mathbf{B}}_1$ controls the emphasis on the current input and historical data. Since these parameters are data-independent, one can interpret them as a global feature of the system. In complex systems (e.g., human neural activity), however, the emphasis depends on the current input, requiring these parameters to be the function of the input (see §4.1). The input-dependency of parameters allows the model to select relevant and filter irrelevant information for each input data, providing a similar mechanism as transformers [33]. Additionally, as we argue earlier, depending on the data, the model needs to adaptively learn if mixing information along the variates is useful. Making parameters input-dependent further overcomes this challenge and lets our model to mix relevant and filter irrelevant variates for the modeling of a variate of interest. One of our main technical contributions is to let $\bar{\mathbf{B}}_i$, $\bar{\mathbf{C}}_i$, and $\Delta_i$ for $i \in \{1, 2\}$ be the function of the input $\mathbf{x}_{v,t}$. This input-dependent 2D SSM, unfortunately, does not have the convolution form, limiting the scalability and efficiency of the training. We overcome this challenge by computing the model recurrently with a new 2D scan.

**2D Selective Scan.** Inspired by the scanning in 1D SSMs [50, 33], we present an algorithm to decrease the sequential steps that are required to calculate hidden states. Given $p, q$, each of which with 6 elements, we first define operation $\circledast$ as: ($\odot$ is matrix-matrix and $\otimes$ is matrix-vector multiplication)

$$p \circledast q = \begin{pmatrix} p_1 & p_2 & p_3 \\ p_4 & p_5 & p_6 \end{pmatrix} \circledast \begin{pmatrix} q_1 & q_2 & q_3 \\ q_4 & q_5 & q_6 \end{pmatrix} = \begin{pmatrix} q_1 \odot p_1 & q_2 \odot p_2 & q_1 \otimes p_3 + q_2 \otimes p_6 + q_3 \\ q_4 \odot p_4 & q_5 \odot p_5 & q_4 \otimes p_3 + q_5 \otimes p_6 + q_6 \end{pmatrix}$$

The proofs of the next two theorems are in Appendix G.

**Theorem 3.2.** *Operator $\circledast$ is associative: Given $p, q$, and $r$, we have: $(p \circledast q) \circledast r = p \circledast (q \circledast r)$.*

**Theorem 3.3.** *2D SSM recurrence can be done in parallel using parallel prefix sum algorithms with associative operator $\circledast$, when fixing the variate.*

### 3.2 New Variant of 2D SSM: 2D Mamba

Figure 2 (Top-Left) shows the recurrence form of our 2D SSM. Each small square is a state of the system, i.e., the state of a variate at a certain time stamp. 2D SSM considers two hidden states for each state (represented by two colors: light red and blue), encoding the information along the time (red) and variate (blue), respectively. Furthermore, each arrow represents a transition matrix $\mathbf{A}_i$ that decides to how information need to be fused. In this section, we discuss a spacial instance of our 2D SSM by limiting its parameters.

**2D Mamba.** We let $\mathbf{A}_2 = \mathbf{A}_3 = \mathbf{0}$ in our 2D SSM. The resulting model is equivalent to:

$$h_{v,t+1}^{(1)} = \bar{\mathbf{A}}_1 h_{v,t}^{(1)} + \bar{\mathbf{B}}_1 \mathbf{x}_{v,t+1}, \tag{10}$$

$$h_{v+1,t}^{(2)} = \bar{\mathbf{A}}_4 h_{v,t}^{(2)} + \bar{\mathbf{B}}_2 \mathbf{x}_{v+1,t}, \tag{11}$$

$$\mathbf{y}_{v,t} = \mathbf{C}_1 h_{v,t}^{(1)} + \mathbf{C}_2 h_{v,t}^{(2)}, \tag{12}$$

where $\bar{\mathbf{A}}_1 = \exp(\Delta_1 \mathbf{A}_1)$, $\bar{\mathbf{A}}_2 = \exp(\Delta_2 \mathbf{A}_2)$, $\bar{\mathbf{B}}_1 = \begin{bmatrix} \mathbf{A}_1^{-1} (\bar{\mathbf{A}}_1 - \mathbf{I}) \mathbf{B}_1^{(1)} \\ \mathbf{0} \end{bmatrix}$, and $\bar{\mathbf{B}}_2 = \begin{bmatrix} \mathbf{0} \\ \mathbf{A}_4^{-1} (\bar{\mathbf{A}}_4 - \mathbf{I}) \mathbf{B}_2^{(2)} \end{bmatrix}$. This formulation with data-dependent parameters, is equivalent to using two

S6 blocks [33] each of which along a dimension. Notably, these two S6 blocks are not separate as the output $\mathbf{y}_{v,t}$ is based on both hidden states $h_{v,t}^{(1)}$ and $h_{v,t}^{(2)}$, capturing 2D inductive bias.

## 3.3 Chimera Neural Architecture

In this section, we use a stack of our 2D SSMs, with non-linearity in between, to enhance the expressive power and capabilities of the abovementioned 2D SSM. To this end, similar to deep SSM models [28], we allow all parameters to be learnable and in each layer we use multiple 2D SSMs, each of which with its own responsibility. Also, in the data-dependent variant of Chimera, we let parameters $\mathbf{B}_i, \mathbf{C}_i$, and $\Delta_i$ for $i \in \{1, 2\}$ be the function of the input $\mathbf{x}$:

$$\mathbf{B}_i = \mathtt{Linear}_{\mathbf{B}_i}(x), \qquad \mathbf{C}_i = \mathtt{Linear}_{\mathbf{C}_i}(x), \qquad \Delta_i = \mathtt{Softplus}\left(\mathtt{Linear}_{\Delta_i}(x)\right). \tag{13}$$

Chimera follows the commonly used decomposition of time series, and decomposes them into trend components and seasonal patterns. it, however, uses special traits of 2D SSM to capture these terms.

**Seasonal Patterns.** To capture the multi-resolution seasonal patterns, we take advantage of the discretization process. Proposition 3.1 states that if $\mathbf{x}(v, t) \mapsto \mathbf{y}(v, t)$ with parameters $(\{\bar{\mathbf{A}}_i\}, \{\bar{\mathbf{B}}_i\}, \{\bar{\mathbf{C}}_i\}, \Delta_1, \Delta_2)$ then $\mathbf{x}(v, kt) \mapsto \mathbf{y}(v, kt)$ with $(\{\bar{\mathbf{A}}_i\}, \{\bar{\mathbf{B}}_i\}, \{\bar{\mathbf{C}}_i\}, k\Delta_1, \Delta_2)$. Accordingly, we use 2D-SSM(.) module with a separate learnable $\Delta_s$ that is responsible to learn the best resolution to capture seasonal patterns. Another interpretation for this module is based on SAR($p, q, s$) (Equation 4). In this case, $\Delta_s$ aims to learn a proper parameter $s$ to capture seasonal patterns. Since we expect the resolution before and after this module matches, we add additional re-discretization module (a simple linear layer), after this module.

**Trend Components.** The second module of Chimera, 2D-SSM$_t$ (.) simply uses a sequence of multiple 2D SSMs to learn trend components. Proper combination of the outputs of this and the previous modules can capture both seasonal and trend components.

**Both Modules Together.** We followed previous studies [51] and consider residual connection modeling for learning trend and seasonal patterns. Given input data $\tilde{\mathbf{X}}_0 = \mathbf{X}$, and $\ell = 0, \ldots, \mathcal{L}$, we have:

$$\hat{\mathbf{X}}_{\ell+1} = \mathtt{2D\text{-}SSM}_t\left(\tilde{\mathbf{X}}_\ell\right), \tag{14}$$

$$\tilde{\mathbf{X}}_{\ell+1} = \mathtt{Re\text{-}Discretization}\left(\mathtt{2D\text{-}SSM}_s\left(\tilde{\mathbf{X}}_\ell - \hat{\mathbf{X}}_{\ell+1}\right)\right). \tag{15}$$

Figure 1 illustrate the architecture of Chimera. Due to the ability of our 2D SSM to recover smoothing techniques (see Theorem 3.4), this combination of modules for trend and seasonal patterns can be viewed as a generalization of traditional methods that use moving average with residual connection to model seasonality [51].

**Gating with Linear Mapping.** Inspired by the success of gated recurrent and SSM-based models [52, 33], we use a head of a fully connected layer with Swish [53], resulting in SwiGLU variant [54]. While we validate the significance of this head, this

**Closed-Loop 2D SSM Decoder.** To enhance the generalizability and the ability of our model for longer-horizon, we extend the closed-loop decoder module [28], which is similar to autoregression, to multivariate time series. We use distinct processes for the inputs and outputs, using additional matrices $\mathbf{D}_1$ and $\mathbf{D}_2$ in each decoder 2D SSM, we model future input time-steps explicitly:

$$\mathbf{y}_{v,t} = \mathbf{C}_1 h_{v,t}^{(1)} + \mathbf{C}_2 h_{v,t}^{(2)}, \tag{16}$$

$$\mathbf{u}_{v,t} = \mathbf{D}_1 h_{v,t}^{(1)} + \mathbf{D}_2 h_{v,t}^{(2)}, \tag{17}$$

where $\mathbf{u}_{v,t}$ is the next input and $\mathbf{y}_{v,t}$ is the output. Note that the other parts (recurrence) are the same as Equation 6. Figure 2 illustrate the architecture of closed-loop 2D SSM.

## 3.4 Theoretical Justification

In this section, we provide some theoretical evidences for the performance of Chimera. These results are mostly revisiting the theorems by Zhang et al. [28] and Baron et al. [49], and extending them

for Chimera. In the first theorem, we show that Chimera recovers several classic methods, and pre-processing steps as it can recover SpaceTime [28] and additionally because of its design, it can recover SARIMA [15]:

**Theorem 3.4.** *Chimera can represent seasonal autoregressive process, SARIMA [15], Space-Time [28], and so ARIMA [14], and exponential smoothing [16].*

**Theorem 3.5.** *Chimera can represent S4nd [32], TSM2 [31], and TSMixer [30].*

Next theorem compares the expressiveness of Chimera with some existing 2D deep SSMs. Since Chimera can recover 2DSSM [49], it can express full-rank kernels with a constant number of parameters:

**Theorem 3.6.** *Similar to 2DSSM [49], Chimera can express full-rank kernels with $\mathcal{O}(1)$ parameters, while existing deep SSMs [32, 31] require $\mathcal{O}(N)$ parameters to express $N$-rank kernels.*

# 4 Experiments

**Goals and Baselines.** We evaluate Chimera on a wide range of time series tasks. In § 4.1 we compare Chimera with the state-of-the-art general multivariate time series models [8, 9, 18, 20, 21, 24, 26, 27, 31, 55–57] on long-term forecasting and classification tasks. In the next part, we test the performance of Chimera in short-term forecasting. In § 4.1 we perform a case study on human neural activity to classify seen images, which requires capturing complex dynamic dependencies of variates, to test the ability of Chimera in capturing cross-variate information and the significance of data-dependency. In § 4.2, we evaluate the significance of the Chimera's components by performing ablation studies. In § 4.2, we evaluate whether the superior performance of Chimera coincide with its efficiency. Finally, we test the Chimera's generalizability on unseen variates and further evaluate its ability to filter irrelevant context in § 4.3. The details and additional experiments are in Appendix I.

Table 1: Average Performance on long-term forecasting task. The first and second results are highlighted in **red** (bold) and orange (underline). Full results are reported in Appendix I.

| | Chimera (ours) | | TSM2 [2024] | | Simba [2024] | | TCN [2024] | | iTransformer [2024] | | RLinear [2023] | | PatchTST [2023] | | Crossformer [2023] | | TiDE [2023] | | TimesNet [2023] | | DLinear [2023] | |
|---|---|---|---|---|---|---|---|---|---|---|---|---|---|---|---|---|---|---|---|---|---|---|
| | MSE | MAE | MSE | MAE | MSE | MAE | MSE | MAE | MSE | MAE | MSE | MAE | MSE | MAE | MSE | MAE | MSE | MAE | MSE | MAE | MSE | MAE |
| ETTm1 | 0.355 | **0.381** | 0.361 | - | 0.383 | 0.396 | **0.351** | **0.381** | 0.407 | 0.410 | 0.414 | 0.407 | 0.387 | 0.400 | 0.513 | 0.496 | 0.419 | 0.419 | 0.400 | 0.406 | 0.403 | 0.407 |
| ETTm2 | **0.252** | 0.317 | 0.267 | - | 0.271 | 0.327 | 0.253 | **0.314** | 0.288 | 0.332 | 0.286 | 0.327 | 0.281 | 0.326 | 0.757 | 0.610 | 0.358 | 0.404 | 0.291 | 0.333 | 0.350 | 0.401 |
| ETTh1 | 0.408 | 0.425 | **0.403** | - | 0.441 | 0.432 | 0.404 | **0.420** | 0.454 | 0.447 | 0.446 | 0.434 | 0.469 | 0.454 | 0.529 | 0.522 | 0.541 | 0.507 | 0.458 | 0.450 | 0.456 | 0.452 |
| ETTh2 | **0.321** | **0.377** | 0.333 | - | 0.361 | 0.391 | 0.322 | 0.379 | 0.383 | 0.407 | 0.374 | 0.398 | 0.387 | 0.407 | 0.942 | 0.684 | 0.611 | 0.550 | 0.414 | 0.427 | 0.559 | 0.515 |
| ECL | **0.154** | **0.249** | 0.169 | - | 0.185 | 0.274 | 0.156 | 0.253 | 0.178 | 0.270 | 0.219 | 0.298 | 0.205 | 0.290 | 0.244 | 0.334 | 0.251 | 0.344 | 0.192 | 0.295 | 0.212 | 0.300 |
| Exchange | 0.311 | **0.358** | 0.443 | - | - | - | **0.302** | 0.366 | 0.360 | 0.403 | 0.378 | 0.417 | 0.367 | 0.404 | 0.940 | 0.707 | 0.370 | 0.413 | 0.416 | 0.443 | 0.354 | 0.414 |
| Traffic | 0.403 | 0.286 | 0.420 | - | 0.493 | 0.291 | **0.398** | **0.270** | 0.428 | 0.282 | 0.626 | 0.378 | 0.481 | 0.304 | 0.550 | 0.304 | 0.760 | 0.473 | 0.620 | 0.336 | 0.625 | 0.383 |
| Weather | **0.219** | **0.258** | 0.239 | - | 0.255 | 0.280 | 0.224 | 0.264 | 0.258 | 0.278 | 0.272 | 0.291 | 0.259 | 0.281 | 0.259 | 0.315 | 0.271 | 0.320 | 0.259 | 0.287 | 0.265 | 0.317 |
| 1st Count | **4** | **5** | 1 | - | 0 | 0 | 3 | 3 | 0 | 0 | 0 | 0 | 0 | 0 | 0 | 0 | 0 | 0 | 0 | 0 | 0 | 0 |

## 4.1 Main Results: Classification and Forecasting

**Long-Term Forecasting.** We perform experiments in long-term forecasting task on benchmark datasets [6]. Table 1 reports the average of results over different horizons (for the results of each see Table 8). Chimera shows outstanding performance, achieving the best or the second best results in all the datasets and outperforms baselines in 5 out of 8 benchmarks. Notably, it surpasses extensively studied MLP-based and Transformer-based models while being more efficient (see Table 3, Figure 4, and Appendix I), providing a better balance of performance and efficiency. It further *significantly* outperforms recurrent models, including very recent Mamba-based architectures [31, 57], unleashing the potential of classical models, SSMs, when are carefully designed in deep learning settings.

**Classification and Anomaly Detection.** We evaluate the performance of Chimera in ECG classification on PTB-XL dataset [7] (see Table 2), speech classification [39] (Table 3), 10 multivariate datasets from UEA Time Series Classification Archive [60] (see Figure 3 and Table 10), and anomaly detection tasks on five widely-used benchmarks: SMD [10], SWaT [61], PSM [62] and SMAP [63] (see Figure 3 and Table 11). For each benchmark, we use the state-of-the-art methods that are

applicable to the task as the baselines. Table 2 reports the performance of Chimera and baselines on ECG classification tasks. Chimera outperforms all the baselines in 4/6 tasks, while achieving the second best results on the remaining tasks. Since these tasks are univariate time series, we attribute the outstanding performance of Chimera, specifically compared to SpaceTime [28], to its ability of capturing seasonal patterns and its input-dependent parameters, resulting in dynamically learn dependencies.

Table 3 reports the results on speech audio classification task, which require long-range modeling of time series. Due to the length of the sequence (16K), LSSL [34] and Transformer [17] has out of memory (OOM) issue, showing the efficiency of Chimera compared to alternative backbones.

Finally, we report the summary of the results in multivariate time series classification and anomaly detection tasks in Figure 3. The full list of results can be found in Table 10 and Table 11. Chimera shows outstanding performance, achieving highest average accuracy and F1 score in classification and anomaly detection tasks even compared to very recent and state-of-the-art methods [8, 9].

Table 2: ECG statement classification on PTB-XL (100 Hz version).

| Tasks | All | Diag | Sub-diag | Super-diag | Form | Rhythm |
|---|---|---|---|---|---|---|
| Chimera | **0.941** | **0.947** | **0.935** | 0.930 | **0.901** | 0.975 |
| SpaceTime [28] | 0.936 | 0.941 | 0.933 | 0.929 | 0.883 | 0.967 |
| S4 [34] | 0.938 | 0.939 | 0.929 | **0.931** | 0.895 | **0.977** |
| Inception | 0.925 | 0.931 | 0.930 | 0.921 | 0.899 | 0.953 |
| xRN-101 | 0.925 | 0.937 | 0.929 | 0.928 | 0.896 | 0.957 |
| LSTM | 0.907 | 0.927 | 0.928 | 0.927 | 0.851 | 0.953 |
| Transformer | 0.857 | 0.876 | 0.882 | 0.887 | 0.771 | 0.831 |

Table 3: Speech classification.

| Method | Acc. (%) |
|---|---|
| Chimera | **98.40** |
| SpaceTime | 97.29 |
| S4 | 98.32 |
| LSSL | OOM |
| WaveGan-D | 96.25 |
| Transformer | OOM |

**Short-Term Forecasting.** Our evaluation on short-term forecasting tasks on M4 benchmark datasets [64] reports in Table 4 (Full list in Table 9), which also shows the superior performance of Chimera compared to baselines.

Table 4: Short-term forecasting task on the M4 dataset. Full results are reported in Appendix I.

| | Models | Chimera (ours) | ModernTCN [2024] | PatchTST [2023] | TimesNet [2023] | N-HiTS [2022] | N-BEATS* [2019] | ETS* [2022] | LightTS [2022] | DLinear [2023] | FED* [2022] | Stationary [2022] | Auto* [2021] | Pyra* [2021] | In* [2021] | Re* [2020] | LSTM [1997] |
|---|---|---|---|---|---|---|---|---|---|---|---|---|---|---|---|---|---|
| Weighted Average | SMAPE | **11.618** | 11.698 | 11.807 | 11.829 | 11.927 | 11.851 | 14.718 | 13.525 | 13.639 | 12.840 | 12.780 | 12.909 | 16.987 | 14.086 | 18.200 | 160.031 |
| | MASE | **1.528** | 1.556 | 1.590 | 1.585 | 1.613 | 1.599 | 2.408 | 2.111 | 2.095 | 1.701 | 1.756 | 1.771 | 3.265 | 2.718 | 4.223 | 25.788 |
| | OWA | **0.827** | 0.838 | 0.851 | 0.851 | 0.861 | 0.855 | 1.172 | 1.051 | 1.051 | 0.918 | 0.930 | 0.939 | 1.480 | 1.230 | 1.775 | 12.642 |

**Case Study of Brain Activity.** Input dependency is a must to capture the dynamic of dependencies. To support this claim, we use BVFC [2] (multivariate time series only), which aim to classify seen images by its corresponding brain activity response. This task, requires focusing more on the dependencies of brain units and their responses rather than the actual time series. Also, since each window corresponds to a specific image, the model needs to capture the dependencies based on the current window, requiring to be input-dependent. Results are reported in Table 5. Chimera significantly outperforms all the baselines including our Chimera but without data-dependent parameters (convolution form). Due to the large number of brain units, i.e., 9K, in the first dataset, transformer-based methods face OOM issue. However, they are also data-dependent and so shows the second best results in second and third datasets. This results support the significance of data-dependency in Chimera.

## 4.2 Ablation Study and Efficiency

To evaluate the significance of the Chimera's design, we perform ablation studies and remove one of the components at each time, keeping other parts unchanged. Table 6 reports the results. The first row reports the Chimera's performance, while row 2 uses unidirectional recurrence along the variate dimension, row 3 removes the gating mechanism, row 4 uses convolution form (data-independent), and row 5 removes the module for seasonal patterns. The results show that all the components of Chimera contributes to its performance.

Table 5: Image classification by brain activity (Acc. %).

| Method | Chimera (ours) | Chimera (ind.) (ours) | SpaceTime [2023] | S4 [2022] | iTrans. [2024] | Trans. [2017] | DLinear [2023] |
|---|---|---|---|---|---|---|---|
| BVFC (9K) | **69.41** | 62.36 | 41.20 | 40.89 | OOM | OOM | 39.74 |
| BVFC (1K) | **58.99** | 50.25 | 34.31 | 35.19 | 54.18 | 43.60 | 33.09 |
| BVFC (400) | **51.08** | 45.17 | 33.58 | 33.76 | 48.22 | 38.05 | 32.73 |

Table 6: Ablation study on the Chimera's design.

| Method | ETTh1 | | ETTm1 | | ETTh2 | |
|---|---|---|---|---|---|---|
| | MSE | MAE | MSE | MAE | MSE | MAE |
| Chimera | 0.408 | 0.425 | 0.355 | 0.381 | 0.321 | 0.377 |
| Uni.-directional | 0.416 | 0.430 | 0.368 | 0.397 | 0.334 | 0.386 |
| w/o Gating | 0.424 | 0.438 | 0.367 | 0.395 | 0.331 | 0.382 |
| Input-independent | 0.473 | 0.501 | 0.433 | 0.412 | 0.380 | 0.414 |
| w/o seasonal | 0.436 | 0.435 | 0.366 | 0.391 | 0.343 | 0.395 |

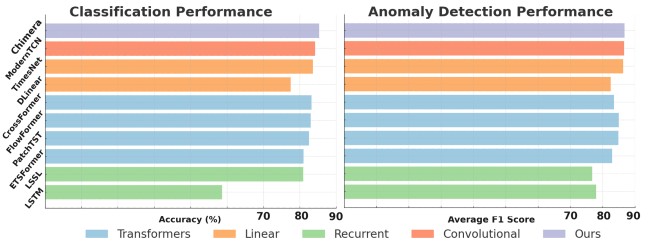

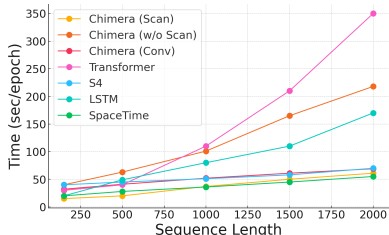

Figure 3: Classification and anomaly detection performance. Full list with additional baselines is in Appendix I.

Figure 4: Wall-clock scaling.

**Length of Time Series.** We perform experiments on the effect of the sequence length on the efficiency of Chimera and baselines. The results are reported in Figure 4. Chimera scales linearly with respect to the sequence length and has smoother scaling than S4 [34] and Transformers [17]. These results also highlight the significance of our algorithm that uses 2D parallel scans for training Chimera. This algorithm results in $\approx \times 4$ faster training, which is very closed to the convolutional format *without* data dependency. Chimera also has a close running time to SpaceTime [28], which has 1D recurrent.

## 4.3 Selection Mechanism Along Time and Variate

**Variate Generalization.** We argue that the data-dependency with discretization allows the model to filter the irrelevant context based on the input, resulting in more generalizability. Inspired by Liu et al. [27], we train our model (and baseline) on 20% of variates and evaluate its generalizability to unseen variates. The results are reported in Figure 5. Chimera has on par generalizability compared to Transformers (when applied along the variate dimension), which we attributes to its data-dependent parameters as Chimera with convolution form performs poorly on unseen variates.

**Context Filtering.** Increasing the lookback length does not necessarily result in better performance for Transformers [27]. Due to the selection mechanism of Chimera, we expect it to filter irrelevant information and monotonically performs better. Figure 6 reports the Chimera's performance (w/ and w/o data-dependency) and transformer-based baselines [6, 22] while varying the lookback length. Chimera due to its selection mechanism monotonically performs better with increasing the lookback.

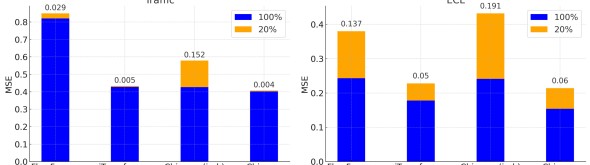

Figure 5: Selection results in generalization to unseen variates.

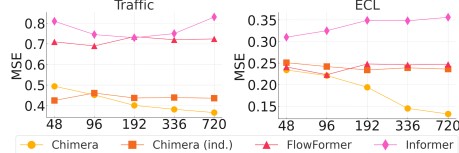

Figure 6: Effect of lookback length.

## 5 Conclusion

This paper presents Chimera, a three-headed 2-dimensional SSM model with provably high expressive power. Chimera is based on 2D SSMs with careful design of parameters that allows it to dynamically and simultaneously capture the dependencies along both time and variate dimensions. We provide different views of our 2D SSM for efficient training, and present a data-dependent formulation with a fast implementation using 2D scans. Our experimental and theoretical results support the effectiveness and efficiency of Chimera in a wide range of tasks.

## Acknowledgments and Disclosure of Funding

This material is based upon work supported by the National Science Foundation under Award No. 2306556. Any opinions, findings and conclusions or recommendations expressed in this material are those of the authors and do not necessarily reflect the views of the National Science Foundation.

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

# A  Background

## A.1   1D Space State Models

1D Space State Models (SSMs) are linear time-invariant systems that map input sequence $x(t) \in \mathbb{R}^L \mapsto y(t) \in \mathbb{R}^L$ [13]. SSMs use a latent state $h(t) \in \mathbb{R}^{N \times L}$, transition parameter $\mathbf{A} \in \mathbb{R}^{N \times N}$, and projection parameters $\mathbf{B} \in \mathbb{R}^{N \times 1}, \mathbf{C} \in \mathbb{R}^{1 \times N}$ to model the input and output as:

$$h'(t) = \mathbf{A}\, h(t) + \mathbf{B}\, x(t), \qquad y(t) = \mathbf{C}\, h(t). \tag{18}$$

Most existing SSMs [34, 33, 31], first discretize the signals $\mathbf{A}, \mathbf{B}$, and $\mathbf{C}$. That is, using a parameter $\boldsymbol{\Delta}$ and zero-order hold, the discretized formulation is defined as:

$$h_t = \bar{\mathbf{A}}\, h_{t-1} + \bar{\mathbf{B}}\, x_t, \qquad y_t = \mathbf{C}\, h_t, \tag{19}$$

where $\bar{\mathbf{A}} = \exp\left(\boldsymbol{\Delta}\mathbf{A}\right)$ and $\bar{\mathbf{B}} = \left(\boldsymbol{\Delta}\mathbf{A}\right)^{-1}\left(\exp\left(\boldsymbol{\Delta}\mathbf{A} - I\right)\right).\boldsymbol{\Delta}\mathbf{B}$. [70] show that discrete SSMs can be interpreted as both convolutions and recurrent networks: i.e.,

$$\bar{\mathbf{K}} = \left(\mathbf{C}\bar{\mathbf{B}}, \mathbf{C}\bar{\mathbf{A}}\bar{\mathbf{B}}, \ldots, \mathbf{C}\bar{\mathbf{A}}^{L-1}\bar{\mathbf{B}}\right),$$
$$y = x * \bar{\mathbf{K}}, \tag{20}$$

which makes their training and inference very efficient as a convolution and recurrent model, respectively.

## A.2   Data Dependency

Above discrete SSMs are based on data-independent parameters. That is, parameters $\boldsymbol{\Delta}, \bar{\mathbf{A}}, \bar{\mathbf{B}}$, and $\mathbf{C}$ are time invariant and are the same for any input. Gu and Dao [33] argue that this time invariance has the cost of limiting SSMs effectiveness in compressing context into a smaller state [33]. To overcome this challenge, they present a selective SSMs (S6) block that effectively selects relevant context by enabling dependence of the parameters $\bar{\mathbf{B}}, \bar{\mathbf{C}}$, and $\boldsymbol{\Delta}$ on the input $x_t$, i.e.:

$$\bar{\mathbf{B}}_t = \texttt{Linear}_{\mathbf{B}}(x_t) \tag{21}$$
$$\bar{\mathbf{C}}_t = \texttt{Linear}_{\mathbf{C}}(x_t) \tag{22}$$
$$\boldsymbol{\Delta}_t = \texttt{Softplus}\left(\texttt{Linear}_{\boldsymbol{\Delta}}(x_t)\right), \tag{23}$$

where $\texttt{Linear}(.)$ is a linear projection and $\texttt{Softplus}(.) = \log(1 + \exp(.))$. This data dependency comes at the cost of efficiency as the model cannot be trained as a convolution. To overcome this challenge, Gu and Dao [33] show that the linear recurrence in Equation 1 can be formulated as an associative scan [71], which accepts efficient parallel algorithms.

# B  Additional Related Work

**Classical Approach.** Modeling time series data is a long-standing problem and has attracted much attention during the past 60 years. There have been several mathematical models to capture the time series traits like exponential smoothing[16], autoregressive integrated moving average (ARIMA) [14], SARIMA [15], Box-Jenkins method [72], and more recently state-space models [12, 13]. Despite their more interpretability, these methods usually fail to capture non-linear dependencies and also often require manually analyzing time series features (e.g., trend or seasonality), resulting in lack of generalizability.

**Recurrent and Deep State Space Models.** Another group of relevant studies to ours is deep sequence models. A common class of architectures for sequence modeling are recurrent neural networks such as like GRUs [73], DeepAR [74], LSTMs [69]. The main drawback of RNNs is their potential for vanishing/exploding gradients and also their slow training. Recently, linear attention methods with fast training attracted attention [75–77]. Katharopoulos et al. [76] show that these methods have recurrent formulation and can be fast in inference.

Recently, deep state space models have attracted much attention as the alternative of Transformers [17], due to their fast training and inference [70]. These methods are the combination of traditional SSMs

with deep neural networks by directly parameterizing the layers of a neural network with multiple linear SSMs, and overcome common recurrent training drawbacks by leveraging the convolutional view of SSMs [70, 45, 46, 34, 50]. Recently, Gu and Dao [33] present a new formulation of deep SSMs by allowing the parameters to be the function of inputs. This architecture shows promising potential in various domains like NLP [33], vision [78, 79, 31], graphs [48], DNA modeling [33, 80].

All the above methods are design for 1D data, meaning that the states depends on one variable. There are, however, a few studies that uses 2D SSMs in deep learning settings. S4ND [32] uses continuous signals to model images. These methods not only consider two separate SSM for the axes, but it also directly treat the system as a continuous system without discretization step. Furthermore, S4ND has data-independent parameters. Another similar approach is 2DSSM [49], that models images as discrete signals. That is, the initial SSM model is discrete and again there is a lack of discretization step, which is important for time series as we discussed earlier. Also, their method again is based on data-independent parameters. Both S4ND and 2DSSM can be computed as a convolution. We, however, present a new scanning technique for fast training of 2D SSMs, even with input-dependent parameters.

WITRAN [81] is a 2D RNN approach for *univariate* time series, which is different from our approaches from different aspects: (1) Goal: The main goal of WITRAN is to use 2D RNN to model univariate time series data. That is, the first dimension corresponds to short-term, and the second dimension corresponds to long-term patterns. On the other hand, our 2D SSM aims to model multivariate time series where the first dimension corresponds to time, and the second dimension corresponds to variates. (2) Backbone: WITRAN is based on GSC (LSTM-like cell), which uses non-linear transition. Our 2D SSM is based on state space models, is simpler, and uses linear transitions. (3) Data-dependency: WITRAN is based on data-independent parameters, meaning that it uses the same transition process for all time steps. 2D SSM, however, is based on data-dependent parameters, which allows it to filter irrelevant/noisy time steps. (4) Recurrence, Hidden States, and Training: Although both models have recurrent form, WITRAN's recurrence is over extracted short-term and long-term patterns. Our 2D SSM's recurrence, however, is over time and variate dimensions. WITRAN uses only one hidden state for each state of the system, while 2D SSM uses 2 different hidden states for each state of the system, allowing more flexibility to capture cross-variate and cross-time information.

**Other methods.** Transformer-based models have attracted much attention over recent years for multivariate time series forecasting, when modeling the complex relationships of co-variates or along the time dimension is required [24–26, 11, 6, 23, 21, 82, 29]. Several studies have focused on designing more efficient and effective attentions with using special traits of time series [20]. Some other studies have focused on extracting long-term information for better forecasting [29, 83]. In addition to transformers, linear models also have shown promising results [8, 30]. For example, Chen et al. [30] present TSMixer, an all-MLP architecture for time series forecasting, with promising performance. Due to the expressive power of our 2D SSM, these linear methods sometimes can be viewed as a special case of 2D SSMs. Recently, convolution-based models for time series have shown promising results [9]. These methods by using global kernels enhance the global receptive field. Our data-independent formulation of Chimera is connected to this line of work as it can be written as a global convolution.

Recently, Koopman-based deep models [84, 85] are designed to capture non-linear dependencies in the data. Existing methods, however, does not consider dynamics in different variates and might miss complex dependencies of variates. Moreover, despite their promising results in the time series forecasting, these approaches can be hard to be parallelizable and so might have longer training for large models.

# C   Details of the Discretization

Given PDE with initial condition $h(0,0) = 0$:

$$\frac{\partial}{\partial t^{(1)}} h^{(1)}\left(t^{(1)}, t^{(2)}\right) = \left(\mathbf{A}_1 h^{(1)}\left(t^{(1)}, t^{(2)}\right), \mathbf{A}_2 h^{(2)}\left(t^{(1)}, t^{(2)}\right)\right) + \mathbf{B}_1 \mathbf{x}\left(t^{(1)}, t^{(2)}\right), \quad (24)$$

$$\frac{\partial}{\partial t^{(1)}} h^{(2)}\left(t^{(1)}, t^{(2)}\right) = \left(\mathbf{A}_1 h^{(1)}\left(t^{(1)}, t^{(2)}\right), \mathbf{A}_2 h^{(2)}\left(t^{(1)}, t^{(2)}\right)\right) + \mathbf{B}_1 \mathbf{x}\left(t^{(1)}, t^{(2)}\right), \quad (25)$$

$$\frac{\partial}{\partial t^{(2)}} h^{(1)}\left(t^{(1)}, t^{(2)}\right) = \left(\mathbf{A}_3 h^{(1)}\left(t^{(1)}, t^{(2)}\right), \mathbf{A}_4 h^{(2)}\left(t^{(1)}, t^{(2)}\right)\right) + \mathbf{B}_2 \mathbf{x}\left(t^{(1)}, t^{(2)}\right), \quad (26)$$

$$\frac{\partial}{\partial t^{(2)}} h^{(2)}\left(t^{(1)}, t^{(2)}\right) = \left(\mathbf{A}_3 h^{(1)}\left(t^{(1)}, t^{(2)}\right), \mathbf{A}_4 h^{(2)}\left(t^{(1)}, t^{(2)}\right)\right) + \mathbf{B}_2 \mathbf{x}\left(t^{(1)}, t^{(2)}\right), \quad (27)$$

over the sampling intervals $[k\Delta t^{(1)}, (k+1)\Delta t^{(1)}]$ and $[\ell\Delta t^{(2)}, (\ell+1)\Delta t^{(2)}]$ we have:

$$\int_{k\Delta t^{(1)}}^{(k+1)\Delta t^{(1)}} \frac{\partial}{\partial t^{(1)}} h^{(1)}\left(t^{(1)}, t^{(2)}\right) dt^{(1)}$$

$$= \int_{k\Delta t^{(1)}}^{(k+1)\Delta t^{(1)}} \left(\mathbf{A}_1 h^{(1)}\left(t^{(1)}, t^{(2)}\right) + \mathbf{B}_1^{(1)} \mathbf{x}^{(1)}\left(t^{(1)}, t^{(2)}\right)\right) dt^{(1)} \qquad (28)$$

and so:

$$\int_{k\Delta t^{(1)}}^{(k+1)\Delta t^{(1)}} \frac{\partial}{\partial t^{(1)}} h^{(2)}\left(t^{(1)}, t^{(2)}\right) dt^{(1)}$$

$$= \int_{k\Delta t^{(1)}}^{(k+1)\Delta t^{(1)}} \left(\mathbf{A}_2 h^{(2)}\left(t^{(1)}, t^{(2)}\right) + \mathbf{B}_1^{(2)} \mathbf{x}^{(2)}\left(t^{(1)}, t^{(2)}\right)\right) dt^{(1)} \qquad (29)$$

Similarly, for the second equation we have:

$$\int_{\ell\Delta t^{(2)}}^{(\ell+1)\Delta t^{(2)}} \frac{\partial}{\partial t^{(2)}} h^{(1)}\left(t^{(1)}, t^{(2)}\right) dt^{(2)}$$

$$= \int_{\ell\Delta t^{(2)}}^{(\ell+1)\Delta t^{(2)}} \left(\mathbf{A}_3 h^{(1)}\left(t^{(1)}, t^{(2)}\right) + \mathbf{B}_2^{(1)} \mathbf{x}^{(1)}\left(t^{(1)}, t^{(2)}\right)\right) dt^{(2)} \qquad (30)$$

and so:

$$\int_{\ell\Delta t^{(2)}}^{(\ell+1)\Delta t^{(2)}} \frac{\partial}{\partial t^{(2)}} h^{(2)}\left(t^{(1)}, t^{(2)}\right) dt^{(2)}$$

$$= \int_{\ell\Delta t^{(2)}}^{(\ell+1)\Delta t^{(2)}} \left(\mathbf{A}_4 h^{(2)}\left(t^{(1)}, t^{(2)}\right) + \mathbf{B}_2^{(2)} \mathbf{x}^{(2)}\left(t^{(1)}, t^{(2)}\right)\right) dt^{(2)} \qquad (31)$$

Next, the integrals can be simplified as:

$$h^{(1)}\left((k+1)\Delta t^{(1)}, t^{(2)}\right)$$

$$= e^{\mathbf{A}_1 \Delta t^{(1)}} h^{(1)}\left(k\Delta t^{(1)}, t^{(2)}\right) + \int_{k\Delta t^{(1)}}^{(k+1)\Delta t^{(1)}} e^{\mathbf{A}_1(t^{(1)} - k\Delta t^{(1)})} \mathbf{B}_1^{(1)} \mathbf{x}^{(1)}\left(t^{(1)}, t^{(2)}\right) dt^{(1)}, \quad (32)$$

and

$$h^{(2)}\left((k+1)\Delta t^{(1)}, t^{(2)}\right)$$

$$= e^{\mathbf{A}_2 \Delta t^{(1)}} h^{(2)}\left(k\Delta t^{(1)}, t^{(2)}\right) + \int_{k\Delta t^{(1)}}^{(k+1)\Delta t^{(1)}} e^{\mathbf{A}_2(t^{(1)} - k\Delta t^{(1)})} \mathbf{B}_1^{(2)} \mathbf{x}^{(2)}\left(t^{(1)}, t^{(2)}\right) dt^{(1)}, \quad (33)$$

and similarly for the third and fourth equations we have:

$$h^{(1)}\left(t^{(1)}, (\ell+1)\Delta t^{(2)}\right)$$

$$= e^{\mathbf{A}_3 \Delta t^{(2)}} h^{(1)}\left(t^{(1)}, \ell\Delta t^{(2)}\right) + \int_{\ell\Delta t^{(2)}}^{(\ell+1)\Delta t^{(2)}} e^{\mathbf{A}_3(t^{(2)} - \ell\Delta t^{(2)})} \mathbf{B}_2^{(1)} \mathbf{x}^{(1)}\left(t^{(2)}, t^{(1)}\right) dt^{(2)} \qquad (34)$$

and

$$
h^{(2)} \left( t^{(1)}, (\ell + 1)\Delta t^{(2)} \right)
$$

$$
= e^{\mathbf{A}_4 \Delta t^{(2)}} h^{(2)} \left( t^{(1)}, \ell \Delta t^{(2)} \right) + \int_{\ell \Delta t^{(2)}}^{(\ell+1)\Delta t^{(2)}} e^{\mathbf{A}_4(t^{(2)} - \ell \Delta t^{(2)})} \mathbf{B}_2^{(2)} \mathbf{x}^{(2)} \left( t^{(2)}, t^{(1)} \right) dt^{(2)} \quad (35)
$$

Using ZOH assumption, we have:

$$
\int_0^{\Delta t^{(1)}} e^{\mathbf{A}_1 s} \, ds = \mathbf{A}^{(1)^{-1}} \left( e^{\mathbf{A}_1 \Delta t^{(1)}} - \mathbf{I} \right)
$$

$$
\int_0^{\Delta t^{(1)}} e^{\mathbf{A}_2 s} \, ds = \mathbf{A}^{(2)^{-1}} \left( e^{\mathbf{A}_2 \Delta t^{(1)}} - \mathbf{I} \right) \quad (36)
$$

$$
\int_0^{\Delta t^{(2)}} e^{\mathbf{A}_3 s} \, ds = \mathbf{A}^{(3)^{-1}} \left( e^{\mathbf{A}_3 \Delta t^{(2)}} - \mathbf{I} \right) \quad (37)
$$

$$
\int_0^{\Delta t^{(2)}} e^{\mathbf{A}_4 s} \, ds = \mathbf{A}^{(4)^{-1}} \left( e^{\mathbf{A}_4 \Delta t^{(2)}} - \mathbf{I} \right) \quad (38)
$$

Accordingly, the discretized form is as follows:

$$
h_{k+1,\ell}^{(1)} = e^{\mathbf{A}_1 \Delta t^{(1)}} h_{k,\ell}^{(1)} + \mathbf{A}^{(1)^{-1}} \left( e^{\mathbf{A}_1 \Delta t^{(1)}} - \mathbf{I} \right) \mathbf{B}_1^{(1)} \mathbf{x}_{k+1,\ell}^{(1)} \quad (39)
$$

$$
h_{k+1,\ell}^{(2)} = e^{\mathbf{A}_2 \Delta t^{(1)}} h_{k,\ell}^{(2)} + \mathbf{A}^{(2)^{-1}} \left( e^{\mathbf{A}_2 \Delta t^{(1)}} - \mathbf{I} \right) \mathbf{B}_1^{(2)} \mathbf{x}_{k+1,\ell}^{(2)} \quad (40)
$$

$$
h_{k,\ell+1}^{(1)} = e^{\mathbf{A}_3 \Delta t^{(2)}} h_{k,\ell}^{(1)} + \mathbf{A}^{(3)^{-1}} \left( e^{\mathbf{A}_3 \Delta t^{(2)}} - \mathbf{I} \right) \mathbf{B}_2^{(1)} \mathbf{x}_{k,\ell+1}^{(1)} \quad (41)
$$

$$
h_{k,\ell+1}^{(2)} = e^{\mathbf{A}_4 \Delta t^{(2)}} h_{k,\ell}^{(2)} + \mathbf{A}^{(4)^{-1}} \left( e^{\mathbf{A}_4 \Delta t^{(2)}} - \mathbf{I} \right) \mathbf{B}_2^{(2)} \mathbf{x}_{k,\ell+1}^{(2)}, \quad (42)
$$

which means that:

$$
\bar{\mathbf{A}}_1 = \exp\left( \mathbf{A}_1 \Delta_1 \right), \quad (43)
$$

$$
\bar{\mathbf{A}}_2 = \exp\left( \mathbf{A}_2 \Delta_1 \right), \quad (44)
$$

$$
\bar{\mathbf{A}}_3 = \exp\left( \mathbf{A}_3 \Delta_2 \right), \quad (45)
$$

$$
\bar{\mathbf{A}}_4 = \exp\left( \mathbf{A}_4 \Delta_2 \right), \quad (46)
$$

$$
\quad (47)
$$

and

$$
\bar{\mathbf{B}}_1 = \begin{bmatrix} \mathbf{A}^{(1)^{-1}} \left( e^{\mathbf{A}_1 \Delta_1} - \mathbf{I} \right) \mathbf{B}_1^{(1)} \\ \mathbf{A}^{(2)^{-1}} \left( e^{\mathbf{A}_2 \Delta_1} - \mathbf{I} \right) \mathbf{B}_1^{(2)} \end{bmatrix}, \quad (48)
$$

$$
\bar{\mathbf{B}}_2 = \begin{bmatrix} \mathbf{A}^{(3)^{-1}} \left( e^{\mathbf{A}_3 \Delta_2} - \mathbf{I} \right) \mathbf{B}_2^{(1)} \\ \mathbf{A}^{(4)^{-1}} \left( e^{\mathbf{A}_4 \Delta_2} - \mathbf{I} \right) \mathbf{B}_2^{(2)} \end{bmatrix}. \quad (49)
$$

# D  Details of the Structure of Transition Matrices

**Definition D.1** (Companion Matrix). *A matrix $A \in \mathbb{R}^{N \times N}$ has companion form if it can be written as:*

$$
A = \begin{pmatrix} 0 & 0 & \dots & 0 & a_1 \\ 1 & 0 & \dots & 0 & a_2 \\ 0 & 1 & \dots & 0 & a_3 \\ \vdots & \vdots & \ddots & \vdots & \vdots \\ 0 & 0 & \dots & 0 & a_{N_1} \\ 0 & 0 & \dots & 1 & a_N \end{pmatrix}. \quad (50)
$$

*These matrices can be decomposed into a shift and a low-rank matrix. That is:*

$$
A = \begin{pmatrix} 0 & 0 & \dots & 0 & a_1 \\ 1 & 0 & \dots & 0 & a_2 \\ 0 & 1 & \dots & 0 & a_3 \\ \vdots & \vdots & \ddots & \vdots & \vdots \\ 0 & 0 & \dots & 0 & a_{N_1} \\ 0 & 0 & \dots & 1 & a_N \end{pmatrix} = \underbrace{\begin{pmatrix} 0 & 0 & \dots & 0 & 0 \\ 1 & 0 & \dots & 0 & 0 \\ 0 & 1 & \dots & 0 & 0 \\ \vdots & \vdots & \ddots & \vdots & \vdots \\ 0 & 0 & \dots & 0 & 0 \\ 0 & 0 & \dots & 1 & 0 \end{pmatrix}}_{\text{Shift Matrix}} + \underbrace{\begin{pmatrix} 0 & 0 & \dots & 0 & a_1 \\ 0 & 0 & \dots & 0 & a_2 \\ 0 & 0 & \dots & 0 & a_3 \\ \vdots & \vdots & \ddots & \vdots & \vdots \\ 0 & 0 & \dots & 0 & a_{N_1} \\ 0 & 0 & \dots & 0 & a_N \end{pmatrix}}_{\text{Low-rank Matrix}}. \tag{51}
$$

This formulation can help us to compute the power of $A$ faster in the convolutional form, as discussed by Zhang et al. [28].

## E    Bidirectionality Across Variates

The causal nature of the 2D SSM result in limited information flow along the variate dimension as variate are not ordered. To overcome this challenge, inspired by the bi-directional 1D SSMs [47, 48, 1], we use two different modules for forward and backward pass along the variate dimension:

$$
h_{v,t+1}^{(1)^f} = \bar{\mathbf{A}}_1^f h_{v,t}^{(1)} + \bar{\mathbf{A}}_2^f h_{v,t}^{(2)^f} + \bar{\mathbf{B}}_1^f \mathbf{x}_{v,t+1},
$$

$$
h_{v,t+1}^{(1)^b} = \bar{\mathbf{A}}_1^b h_{v,t}^{(1)} + \bar{\mathbf{A}}_2^b h_{v,t}^{(2)} + \bar{\mathbf{B}}_1^b \mathbf{x}_{v,t+1}, \tag{52}
$$

$$
h_{v+1,t}^{(2)^f} = \bar{\mathbf{A}}_3^f h_{v,t}^{(1)} + \bar{\mathbf{A}}_4^f h_{v,t}^{(2)^f} + \bar{\mathbf{B}}_2^f \mathbf{x}_{v+1,t},
$$

$$
h_{v-1,t}^{(2)^b} = \bar{\mathbf{A}}_3^b h_{v,t}^{(1)^b} + \bar{\mathbf{A}}_4^b h_{v,t}^{(2)^b} + \bar{\mathbf{B}}_2^b \mathbf{x}_{v-1,t}, \tag{53}
$$

$$
\mathbf{y}_{v,t}^f = \mathbf{C}_1^f h_{v,t}^{(1)^f} + \mathbf{C}_2^f h_{v,t}^{(2)^f}, \tag{54}
$$

$$
\mathbf{y}_{v,t}^b = \mathbf{C}_1^b h_{v,t}^{(1)^b} + \mathbf{C}_2^b h_{v,t}^{(2)^b}, \tag{55}
$$

$$
\mathbf{y}_{v,t} = \mathbf{y}_{v,t}^f + \mathbf{y}_{v,t}^b, \tag{56}
$$

where $\bar{\mathbf{A}}_1^\tau, \bar{\mathbf{A}}_2^\tau, \bar{\mathbf{A}}_3^\tau, \bar{\mathbf{A}}_4^\tau \in \mathbb{R}^{N \times N}$, $\bar{\mathbf{B}}_1^\tau, \bar{\mathbf{B}}_2^\tau \in \mathbb{R}^{N \times 1}$, and $\mathbf{C}_1^\tau, \mathbf{C}_2^\tau \in \mathbb{R}^{1 \times N}$ are parameters of the model, $h_{v,t}^{(1)^\tau}, h_{v,t}^{(2)^\tau} \in \mathbb{R}^{N \times d}$ are hidden states, $\mathbf{x}_{v,t} \in \mathbb{R}^{1 \times d}$ is the input, and $\tau \in \{f, b\}$. Figure 2 illustrates the bi-directional recurrence process in Chimera. For the sake of simplicity, we continue with unidirectional pass, but adapting them for bi-directional setting is simple as we use two separate blocks, each of which for a direction.

## F    Time Complexity of 2D-SSM

Let $T$ be the length of the time series and $V$ be the number of variates. Since for each variate we have $\mathcal{O}(T)$ recurrence and we have $V$ variates, the total number of recurrence is $\mathcal{O}(TV)$. In each step, we have the matrix multiplication of transition matrices $A_i$ with the hidden states and also $B_i$s with the input. Accordingly, if we use $d$ as the state dimension, the complexity is $\mathcal{O}(d^2 TV)$. Given the fact that $d$ is usually a small number, the overall complexity is linear with respect to each of $V$ and $T$.

## G    Theoretical Results

### G.1    Proof of Theorem 3.2

In this part, we want to prove that $\circledast$ is associative. This operator is defined as:

$$
p \circledast q = \begin{pmatrix} p_1 & p_2 & p_3 \\ p_4 & p_5 & p_6 \end{pmatrix} \circledast \begin{pmatrix} q_1 & q_2 & q_3 \\ q_4 & q_5 & q_6 \end{pmatrix} = \begin{pmatrix} q_1 \odot p_1 & q_2 \odot p_2 & q_1 \otimes p_3 + q_2 \otimes p_6 + q_3 \\ q_4 \odot p_4 & q_5 \odot p_5 & q_4 \otimes p_3 + q_5 \otimes p_6 + q_6 \end{pmatrix}
$$

Accordingly, we have:

$$
(p \circledast q) \circledast r = \begin{pmatrix} q_1 \odot p_1 & q_2 \odot p_2 & q_1 \otimes p_3 + q_2 \otimes p_6 + q_3 \\ q_4 \odot p_4 & q_5 \odot p_5 & q_4 \otimes p_3 + q_5 \otimes p_6 + q_6 \end{pmatrix} \circledast \begin{pmatrix} r_1 & r_2 & r_3 \\ r_4 & r_5 & r_6 \end{pmatrix}, \tag{57}
$$

Table 7: Dataset descriptions. The dataset size is organized in (Train, Validation, Test).

| Tasks | Dataset | Dim | Series Length | Dataset Size | Information (Frequency) |
|---|---|---|---|---|---|
| Forecasting (Long-term) | ETTm1, ETTm2 | 7 | {96, 192, 336, 720} | (34465, 11521, 11521) | Electricity (15 mins) |
| | ETTh1, ETTh2 | 7 | {96, 192, 336, 720} | (8545, 2881, 2881) | Electricity (15 mins) |
| | Electricity | 321 | {96, 192, 336, 720} | (18317, 2633, 5261) | Electricity (Hourly) |
| | Traffic | 862 | {96, 192, 336, 720} | (12185, 1757, 3509) | Transportation (Hourly) |
| | Weather | 21 | {96, 192, 336, 720} | (36792, 5271, 10540) | Weather (10 mins) |
| | Exchange | 8 | {96, 192, 336, 720} | (5120, 665, 1422) | Exchange rate (Daily) |
| Forecasting (short-term) | M4-Yearly | 1 | 6 | (23000, 0, 23000) | Demographic |
| | M4-Quarterly | 1 | 8 | (24000, 0, 24000) | Finance |
| | M4-Monthly | 1 | 18 | (48000, 0, 48000) | Industry |
| | M4-Weakly | 1 | 13 | (359, 0, 359) | Macro |
| | M4-Daily | 1 | 14 | (4227, 0, 4227) | Micro |
| | M4-Hourly | 1 | 48 | (414, 0, 414) | Other |
| Classification (UEA) | EthanolConcentration | 3 | 1751 | (261, 0, 263) | Alcohol Industry |
| | FaceDetection | 144 | 62 | (5890, 0, 3524) | Face (250Hz) |
| | Handwriting | 3 | 152 | (150, 0, 850) | Handwriting |
| | Heartbeat | 61 | 405 | (204, 0, 205) | Heart Beat |
| | JapaneseVowels | 12 | 29 | (270, 0, 370) | Voice |
| | PEMS-SF | 963 | 144 | (267, 0, 173) | Transportation (Daily) |
| | SelfRegulationSCP1 | 6 | 896 | (268, 0, 293) | Health (256Hz) |
| | SelfRegulationSCP2 | 7 | 1152 | (200, 0, 180) | Health (256Hz) |
| | SpokenArabicDigits | 13 | 93 | (6599, 0, 2199) | Voice (11025Hz) |
| | UWaveGestureLibrary | 3 | 315 | (120, 0, 320) | Gesture |
| Anomaly Detection | SMD | 38 | 100 | (566724, 141681, 708420) | Server Machine |
| | MSL | 55 | 100 | (44653, 11664, 73729) | Spacecraft |
| | SMAP | 25 | 100 | (108146, 27037, 427617) | Spacecraft |
| | SWaT | 51 | 100 | (396000, 99000, 449919) | Infrastructure |
| | PSM | 25 | 100 | (105984, 26497, 87841) | Server Machine |

re-using the definition of $\divideontimes$, we have:

$$(p \divideontimes q) \divideontimes r = \begin{pmatrix} q_1 \odot p_1 & q_2 \odot p_2 & q_1 \otimes p_3 + q_2 \otimes p_6 + q_3 \\ q_4 \odot p_4 & q_5 \odot p_5 & q_4 \otimes p_3 + q_5 \otimes p_6 + q_6 \end{pmatrix} \divideontimes \begin{pmatrix} r_1 & r_2 & r_3 \\ r_4 & r_5 & r_6 \end{pmatrix} \tag{58}$$

$$= \begin{pmatrix} r_1 \odot (q_1 \odot p_1) & r_2 \odot (q_2 \odot p_2) & r_1 \otimes (q_1 \otimes p_3 + q_2 \otimes p_6 + q_3) + r_2 \odot (q_4 \otimes p_3 + q_5 \otimes p_6 + q_6) + r_3 \\ r_4 \odot (q_4 \odot p_4) & r_5 \odot (q_5 \odot p_5) & r_4 \otimes (q_1 \otimes p_3 + q_2 \otimes p_6 + q_3) + r_4 \otimes (q_4 \otimes p_3 + q_5 \otimes p_6 + q_6) + r_6 \end{pmatrix} \tag{59}$$

Table 8: Long-term forecasting task with different horizons $\mathbf{H}$.

| | | Chimera (ours) | | TSM2 [2024] | | Simba [2024] | | TCN [2024] | | iTransformer [2024] | | RLinear [2023] | | PatchTST [2023] | | Crossformer [2023] | | TiDE [2023] | | TimesNet [2023] | | DLinear [2023] | | SCINet [2022] | | FEDformer [2022] | | Stationary [2022] | | Autoformer [2021] | |
|---|---|---|---|---|---|---|---|---|---|---|---|---|---|---|---|---|---|---|---|---|---|---|---|---|---|---|---|---|---|---|---|
| | | MSE | MAE | MSE | MAE | MSE | MAE | MSE | MAE | MSE | MAE | MSE | MAE | MSE | MAE | MSE | MAE | MSE | MAE | MSE | MAE | MSE | MAE | MSE | MAE | MSE | MAE | MSE | MAE | MSE | MAE |
| ETTm1 | 96 | 0.318 | 0.354 | 0.322 | - | 0.324 | 0.360 | 0.292 | 0.346 | 0.334 | 0.368 | 0.355 | 0.376 | 0.329 | 0.367 | 0.404 | 0.426 | 0.364 | 0.387 | 0.338 | 0.375 | 0.345 | 0.372 | 0.418 | 0.438 | 0.379 | 0.419 | 0.386 | 0.398 | 0.505 | 0.475 |
| | 192 | 0.331 | 0.369 | 0.349 | - | 0.363 | 0.382 | 0.332 | 0.368 | 0.377 | 0.391 | 0.391 | 0.392 | 0.367 | 0.385 | 0.450 | 0.451 | 0.398 | 0.404 | 0.374 | 0.387 | 0.380 | 0.389 | 0.439 | 0.450 | 0.426 | 0.441 | 0.459 | 0.444 | 0.553 | 0.496 |
| | 336 | 0.363 | 0.389 | 0.366 | - | 0.395 | 0.405 | 0.365 | 0.391 | 0.426 | 0.420 | 0.424 | 0.415 | 0.399 | 0.410 | 0.532 | 0.515 | 0.428 | 0.425 | 0.410 | 0.411 | 0.413 | 0.413 | 0.490 | 0.485 | 0.445 | 0.459 | 0.495 | 0.464 | 0.621 | 0.537 |
| | 720 | 0.409 | 0.415 | 0.407 | - | 0.451 | 0.437 | 0.416 | 0.417 | 0.491 | 0.459 | 0.487 | 0.450 | 0.454 | 0.439 | 0.666 | 0.589 | 0.487 | 0.461 | 0.478 | 0.450 | 0.474 | 0.453 | 0.595 | 0.550 | 0.543 | 0.490 | 0.585 | 0.516 | 0.671 | 0.561 |
| | Avg | 0.355 | 0.381 | 0.361 | - | 0.383 | 0.396 | 0.351 | 0.381 | 0.407 | 0.410 | 0.414 | 0.407 | 0.387 | 0.400 | 0.513 | 0.496 | 0.419 | 0.419 | 0.400 | 0.406 | 0.403 | 0.407 | 0.485 | 0.481 | 0.448 | 0.452 | 0.481 | 0.456 | 0.588 | 0.517 |
| ETTm2 | 96 | 0.169 | 0.265 | 0.173 | - | 0.177 | 0.263 | 0.166 | 0.256 | 0.180 | 0.264 | 0.182 | 0.265 | 0.175 | 0.259 | 0.287 | 0.366 | 0.207 | 0.305 | 0.187 | 0.267 | 0.193 | 0.292 | 0.286 | 0.377 | 0.203 | 0.287 | 0.192 | 0.274 | 0.255 | 0.339 |
| | 192 | 0.221 | 0.290 | 0.230 | - | 0.245 | 0.306 | 0.222 | 0.293 | 0.250 | 0.309 | 0.246 | 0.304 | 0.241 | 0.302 | 0.414 | 0.492 | 0.290 | 0.364 | 0.249 | 0.309 | 0.284 | 0.362 | 0.399 | 0.445 | 0.269 | 0.328 | 0.280 | 0.339 | 0.281 | 0.340 |
| | 336 | 0.279 | 0.339 | 0.279 | - | 0.304 | 0.343 | 0.272 | 0.324 | 0.311 | 0.348 | 0.307 | 0.342 | 0.305 | 0.343 | 0.597 | 0.542 | 0.377 | 0.422 | 0.321 | 0.351 | 0.369 | 0.427 | 0.637 | 0.591 | 0.325 | 0.366 | 0.334 | 0.361 | 0.339 | 0.372 |
| | 720 | 0.342 | 0.376 | 0.388 | - | 0.400 | 0.399 | 0.351 | 0.381 | 0.412 | 0.407 | 0.407 | 0.398 | 0.402 | 0.400 | 1.730 | 1.042 | 0.558 | 0.524 | 0.408 | 0.403 | 0.554 | 0.522 | 0.960 | 0.735 | 0.421 | 0.415 | 0.417 | 0.413 | 0.433 | 0.432 |
| | Avg | 0.252 | 0.317 | 0.267 | - | 0.271 | 0.327 | 0.253 | 0.314 | 0.288 | 0.332 | 0.286 | 0.327 | 0.281 | 0.326 | 0.757 | 0.610 | 0.358 | 0.404 | 0.291 | 0.333 | 0.350 | 0.401 | 0.571 | 0.537 | 0.305 | 0.349 | 0.306 | 0.347 | 0.327 | 0.371 |
| ETTh1 | 96 | 0.366 | 0.392 | 0.375 | - | 0.379 | 0.395 | 0.368 | 0.394 | 0.386 | 0.405 | 0.386 | 0.395 | 0.414 | 0.419 | 0.423 | 0.448 | 0.479 | 0.464 | 0.384 | 0.402 | 0.386 | 0.400 | 0.654 | 0.599 | 0.376 | 0.419 | 0.513 | 0.491 | 0.449 | 0.459 |
| | 192 | 0.402 | 0.414 | 0.398 | - | 0.432 | 0.424 | 0.405 | 0.413 | 0.441 | 0.436 | 0.437 | 0.424 | 0.460 | 0.445 | 0.471 | 0.474 | 0.525 | 0.492 | 0.436 | 0.429 | 0.437 | 0.432 | 0.719 | 0.631 | 0.420 | 0.448 | 0.534 | 0.504 | 0.500 | 0.482 |
| | 336 | 0.406 | 0.419 | 0.419 | - | 0.473 | 0.443 | 0.391 | 0.412 | 0.487 | 0.458 | 0.479 | 0.446 | 0.501 | 0.466 | 0.570 | 0.546 | 0.565 | 0.515 | 0.491 | 0.469 | 0.481 | 0.459 | 0.778 | 0.659 | 0.459 | 0.465 | 0.588 | 0.535 | 0.521 | 0.496 |
| | 720 | 0.458 | 0.477 | 0.422 | - | 0.483 | 0.469 | 0.450 | 0.461 | 0.503 | 0.491 | 0.481 | 0.470 | 0.500 | 0.488 | 0.653 | 0.621 | 0.594 | 0.558 | 0.521 | 0.500 | 0.519 | 0.516 | 0.836 | 0.699 | 0.506 | 0.507 | 0.643 | 0.616 | 0.514 | 0.512 |
| | Avg | 0.408 | 0.425 | 0.403 | - | 0.441 | 0.432 | 0.404 | 0.420 | 0.454 | 0.447 | 0.446 | 0.434 | 0.469 | 0.454 | 0.529 | 0.522 | 0.541 | 0.507 | 0.458 | 0.450 | 0.456 | 0.452 | 0.747 | 0.647 | 0.440 | 0.460 | 0.570 | 0.537 | 0.496 | 0.487 |
| ETTh2 | 96 | 0.262 | 0.327 | 0.253 | - | 0.290 | 0.339 | 0.263 | 0.332 | 0.297 | 0.349 | 0.288 | 0.338 | 0.302 | 0.348 | 0.745 | 0.584 | 0.400 | 0.440 | 0.340 | 0.374 | 0.333 | 0.387 | 0.707 | 0.621 | 0.358 | 0.397 | 0.476 | 0.458 | 0.346 | 0.388 |
| | 192 | 0.320 | 0.372 | 0.334 | - | 0.373 | 0.390 | 0.320 | 0.374 | 0.380 | 0.400 | 0.374 | 0.390 | 0.388 | 0.400 | 0.877 | 0.656 | 0.528 | 0.509 | 0.402 | 0.414 | 0.477 | 0.476 | 0.860 | 0.689 | 0.429 | 0.439 | 0.512 | 0.493 | 0.456 | 0.452 |
| | 336 | 0.316 | 0.381 | 0.347 | - | 0.376 | 0.406 | 0.313 | 0.376 | 0.428 | 0.432 | 0.415 | 0.426 | 0.426 | 0.433 | 1.043 | 0.731 | 0.643 | 0.571 | 0.452 | 0.452 | 0.594 | 0.541 | 1.000 | 0.744 | 0.496 | 0.487 | 0.552 | 0.551 | 0.482 | 0.486 |
| | 720 | 0.389 | 0.430 | 0.401 | - | 0.407 | 0.431 | 0.392 | 0.433 | 0.427 | 0.445 | 0.420 | 0.440 | 0.431 | 0.446 | 1.104 | 0.763 | 0.874 | 0.679 | 0.462 | 0.468 | 0.831 | 0.657 | 1.249 | 0.838 | 0.463 | 0.474 | 0.562 | 0.560 | 0.515 | 0.511 |
| | Avg | 0.321 | 0.377 | 0.333 | - | 0.361 | 0.391 | 0.322 | 0.379 | 0.383 | 0.407 | 0.374 | 0.398 | 0.387 | 0.407 | 0.942 | 0.684 | 0.611 | 0.550 | 0.414 | 0.427 | 0.559 | 0.515 | 0.954 | 0.723 | 0.437 | 0.449 | 0.526 | 0.516 | 0.450 | 0.459 |
| ECL | 96 | 0.132 | 0.234 | 0.142 | - | 0.165 | 0.253 | 0.129 | 0.226 | 0.148 | 0.240 | 0.201 | 0.281 | 0.181 | 0.270 | 0.219 | 0.314 | 0.237 | 0.329 | 0.168 | 0.272 | 0.197 | 0.282 | 0.247 | 0.345 | 0.193 | 0.308 | 0.169 | 0.273 | 0.201 | 0.317 |
| | 192 | 0.144 | 0.223 | 0.153 | - | 0.173 | 0.262 | 0.143 | 0.239 | 0.162 | 0.253 | 0.201 | 0.283 | 0.188 | 0.274 | 0.231 | 0.322 | 0.236 | 0.330 | 0.184 | 0.289 | 0.196 | 0.285 | 0.257 | 0.355 | 0.201 | 0.315 | 0.182 | 0.286 | 0.222 | 0.334 |
| | 336 | 0.156 | 0.259 | 0.175 | - | 0.188 | 0.277 | 0.161 | 0.259 | 0.178 | 0.269 | 0.215 | 0.298 | 0.204 | 0.293 | 0.246 | 0.337 | 0.249 | 0.344 | 0.198 | 0.300 | 0.209 | 0.301 | 0.269 | 0.369 | 0.214 | 0.329 | 0.200 | 0.304 | 0.231 | 0.338 |
| | 720 | 0.184 | 0.280 | 0.209 | - | 0.214 | 0.305 | 0.191 | 0.286 | 0.225 | 0.317 | 0.257 | 0.331 | 0.246 | 0.324 | 0.280 | 0.363 | 0.284 | 0.373 | 0.220 | 0.320 | 0.245 | 0.333 | 0.299 | 0.390 | 0.246 | 0.355 | 0.222 | 0.321 | 0.254 | 0.361 |
| | Avg | 0.154 | 0.249 | 0.169 | - | 0.185 | 0.274 | 0.156 | 0.253 | 0.178 | 0.270 | 0.219 | 0.298 | 0.205 | 0.290 | 0.244 | 0.334 | 0.251 | 0.344 | 0.192 | 0.295 | 0.212 | 0.300 | 0.268 | 0.365 | 0.214 | 0.327 | 0.193 | 0.296 | 0.227 | 0.338 |
| Exchange | 96 | 0.077 | 0.198 | 0.163 | - | - | - | 0.080 | 0.196 | 0.086 | 0.206 | 0.093 | 0.217 | 0.088 | 0.205 | 0.256 | 0.367 | 0.094 | 0.218 | 0.107 | 0.234 | 0.088 | 0.218 | 0.267 | 0.396 | 0.148 | 0.278 | 0.111 | 0.237 | 0.197 | 0.323 |
| | 192 | 0.159 | 0.270 | 0.229 | - | - | - | 0.166 | 0.288 | 0.177 | 0.299 | 0.184 | 0.307 | 0.176 | 0.299 | 0.470 | 0.509 | 0.184 | 0.307 | 0.226 | 0.344 | 0.176 | 0.315 | 0.351 | 0.459 | 0.271 | 0.315 | 0.219 | 0.335 | 0.300 | 0.369 |
| | 336 | 0.311 | 0.344 | 0.383 | - | - | - | 0.307 | 0.398 | 0.331 | 0.417 | 0.351 | 0.432 | 0.301 | 0.397 | 1.268 | 0.883 | 0.349 | 0.431 | 0.367 | 0.448 | 0.313 | 0.427 | 1.324 | 0.853 | 0.460 | 0.427 | 0.421 | 0.476 | 0.509 | 0.524 |
| | 720 | 0.697 | 0.623 | 0.999 | - | - | - | 0.656 | 0.582 | 0.847 | 0.691 | 0.886 | 0.714 | 0.901 | 0.714 | 1.767 | 1.068 | 0.852 | 0.698 | 0.964 | 0.746 | 0.839 | 0.695 | 1.058 | 0.797 | 1.195 | 0.695 | 1.092 | 0.769 | 1.447 | 0.941 |
| | Avg | 0.311 | 0.358 | 0.443 | - | - | - | 0.302 | 0.366 | 0.360 | 0.403 | 0.378 | 0.417 | 0.367 | 0.404 | 0.940 | 0.707 | 0.370 | 0.413 | 0.416 | 0.443 | 0.354 | 0.414 | 0.750 | 0.626 | 0.519 | 0.429 | 0.461 | 0.454 | 0.613 | 0.539 |
| Traffic | 96 | 0.366 | 0.248 | 0.396 | - | 0.468 | 0.268 | 0.368 | 0.253 | 0.395 | 0.268 | 0.649 | 0.389 | 0.462 | 0.295 | 0.522 | 0.290 | 0.805 | 0.493 | 0.593 | 0.321 | 0.650 | 0.396 | 0.788 | 0.499 | 0.587 | 0.366 | 0.612 | 0.338 | 0.613 | 0.388 |
| | 192 | 0.394 | 0.292 | 0.408 | - | 0.413 | 0.317 | 0.379 | 0.261 | 0.417 | 0.276 | 0.601 | 0.366 | 0.466 | 0.296 | 0.530 | 0.293 | 0.756 | 0.474 | 0.617 | 0.336 | 0.598 | 0.370 | 0.789 | 0.505 | 0.604 | 0.373 | 0.613 | 0.340 | 0.616 | 0.382 |
| | 336 | 0.409 | 0.311 | 0.427 | - | 0.529 | 0.284 | 0.397 | 0.270 | 0.433 | 0.283 | 0.609 | 0.369 | 0.482 | 0.304 | 0.558 | 0.305 | 0.762 | 0.477 | 0.629 | 0.336 | 0.605 | 0.373 | 0.797 | 0.508 | 0.621 | 0.383 | 0.618 | 0.328 | 0.622 | 0.337 |
| | 720 | 0.443 | 0.294 | 0.449 | - | 0.564 | 0.297 | 0.440 | 0.296 | 0.467 | 0.302 | 0.647 | 0.387 | 0.514 | 0.322 | 0.589 | 0.328 | 0.719 | 0.449 | 0.640 | 0.350 | 0.645 | 0.394 | 0.841 | 0.523 | 0.626 | 0.382 | 0.653 | 0.355 | 0.660 | 0.408 |
| | Avg | 0.403 | 0.286 | 0.420 | - | 0.493 | 0.291 | 0.398 | 0.270 | 0.428 | 0.282 | 0.626 | 0.378 | 0.481 | 0.304 | 0.550 | 0.304 | 0.760 | 0.473 | 0.620 | 0.336 | 0.625 | 0.383 | 0.804 | 0.509 | 0.610 | 0.376 | 0.624 | 0.340 | 0.628 | 0.379 |
| Weather | 96 | 0.146 | 0.206 | 0.161 | - | 0.176 | 0.219 | 0.149 | 0.200 | 0.174 | 0.214 | 0.192 | 0.232 | 0.177 | 0.218 | 0.158 | 0.230 | 0.202 | 0.261 | 0.172 | 0.220 | 0.196 | 0.255 | 0.221 | 0.306 | 0.217 | 0.296 | 0.173 | 0.223 | 0.266 | 0.336 |
| | 192 | 0.189 | 0.239 | 0.208 | - | 0.222 | 0.260 | 0.196 | 0.245 | 0.221 | 0.254 | 0.240 | 0.271 | 0.225 | 0.259 | 0.206 | 0.277 | 0.242 | 0.298 | 0.219 | 0.261 | 0.237 | 0.296 | 0.261 | 0.340 | 0.276 | 0.336 | 0.245 | 0.285 | 0.307 | 0.367 |
| | 336 | 0.244 | 0.281 | 0.252 | - | 0.275 | 0.297 | 0.238 | 0.277 | 0.278 | 0.296 | 0.292 | 0.307 | 0.278 | 0.297 | 0.272 | 0.335 | 0.287 | 0.335 | 0.280 | 0.306 | 0.283 | 0.335 | 0.309 | 0.378 | 0.339 | 0.380 | 0.321 | 0.338 | 0.359 | 0.395 |
| | 720 | 0.297 | 0.309 | 0.337 | - | 0.350 | 0.349 | 0.314 | 0.334 | 0.358 | 0.347 | 0.364 | 0.353 | 0.354 | 0.348 | 0.398 | 0.418 | 0.351 | 0.386 | 0.365 | 0.359 | 0.345 | 0.381 | 0.377 | 0.427 | 0.403 | 0.428 | 0.414 | 0.410 | 0.419 | 0.428 |
| | Avg | 0.219 | 0.258 | 0.239 | - | 0.255 | 0.280 | 0.224 | 0.264 | 0.258 | 0.278 | 0.272 | 0.291 | 0.259 | 0.281 | 0.259 | 0.315 | 0.271 | 0.320 | 0.259 | 0.287 | 0.265 | 0.317 | 0.292 | 0.363 | 0.309 | 0.360 | 0.288 | 0.314 | 0.338 | 0.382 |

Using the fact that $\odot$ and $\otimes$ are associative, we have:

$$(p \circledast q) \circledast r = \begin{pmatrix} q_1 \odot p_1 & q_2 \odot p_2 & q_1 \otimes p_3 + q_2 \otimes p_6 + q_3 \\ q_4 \odot p_4 & q_5 \odot p_5 & q_4 \otimes p_3 + q_5 \otimes p_6 + q_6 \end{pmatrix} \circledast \begin{pmatrix} r_1 & r_2 & r_3 \\ r_4 & r_5 & r_6 \end{pmatrix} \tag{60}$$

$$= \begin{pmatrix} r_1 \odot (q_1 \odot p_1) & r_2 \odot (q_2 \odot p_2) & r_1 \otimes (q_1 \otimes p_3 + q_2 \otimes p_6 + q_3) + r_2 \odot (q_4 \otimes p_3 + q_5 \otimes p_6 + q_6) + r_3 \\ r_4 \odot (q_4 \odot p_4) & r_5 \odot (q_5 \odot p_5) & r_4 \otimes (q_1 \otimes p_3 + q_2 \otimes p_6 + q_3) + r_4 \otimes (q_4 \otimes p_3 + q_5 \otimes p_6 + q_6) + r_6 \end{pmatrix} \tag{61}$$

$$= \begin{pmatrix} p_1 & p_2 & p_3 \\ p_4 & p_5 & p_6 \end{pmatrix} \circledast \begin{pmatrix} r_1 \odot q_1 & r_2 \odot q_2 & r_1 \otimes q_3 + r_2 \otimes q_6 + r_3 \\ r_4 \odot q_4 & r_5 \odot q_5 & r_4 \otimes q_3 + r_5 \otimes q_6 + r_6 \end{pmatrix} \tag{62}$$

$$= p \circledast (q \circledast r), \tag{63}$$

which proves the theorem.

## G.2 Proof of Theorem 3.3

For each $v, t$, we can pre-compute $\mathbf{B}_1 \mathbf{x}_{v,t}$ and $\mathbf{B}_2 \mathbf{x}_{v,t+1}$. Accordingly, all the following parameters are pre-computed:

$$c_{v,t}^{(i,j,k,\ell)} = \begin{pmatrix} \mathbf{A}_1 & \mathbf{A}_2 & \mathbf{B}_1 \mathbf{x}_{v+i,t+j} \\ \mathbf{A}_3 & \mathbf{A}_4 & \mathbf{B}_2 \mathbf{x}_{v+k,t+\ell} \end{pmatrix}, \tag{64}$$

Table 9: Full results for the short-term forecasting task in the M4 dataset. ∗. in the Transformers indicates the name of ∗former. *Stationary* means the Non-stationary Transformer.

| Models | | Chimera (ours) | ModernTCN [2024] | PatchTST [2023] | TimesNet [2023] | N-HiTS [2022] | N-BEATS* [2019] | ETS* [2022] | LightTS [2022] | DLinear [2023] | FED* [2022] | Stationary [2022] | Auto* [2021] | Pyra* [2021] | In* [2021] | Re* [2020] | LSTM [1997] |
|---|---|---|---|---|---|---|---|---|---|---|---|---|---|---|---|---|---|
| Yearly | SMAPE | **13.107** | 13.226 | 13.258 | 13.387 | 13.418 | 13.436 | 18.009 | 14.247 | 16.965 | 13.728 | 13.717 | 13.974 | 15.530 | 14.727 | 16.169 | 176.040 |
| | MASE | **2.902** | 2.957 | 2.985 | 2.996 | 3.045 | 3.043 | 4.487 | 3.109 | 4.283 | 3.048 | 3.078 | 3.134 | 3.711 | 3.418 | 3.800 | 31.033 |
| | OWA | **0.767** | 0.777 | 0.781 | 0.786 | 0.793 | 0.794 | 1.115 | 0.827 | 1.058 | 0.803 | 0.807 | 0.822 | 0.942 | 0.881 | 0.973 | 9.290 |
| Quarterly | SMAPE | **9.892** | 9.971 | 10.179 | 10.100 | 10.202 | 10.124 | 13.376 | 11.364 | 12.145 | 10.792 | 10.958 | 11.338 | 15.449 | 11.360 | 13.313 | 172.808 |
| | MASE | 1.105 | 1.167 | **0.803** | 1.182 | 1.194 | 1.169 | 1.906 | 1.328 | 1.520 | 1.283 | 1.325 | 1.365 | 2.350 | 1.401 | 1.775 | 19.753 |
| | OWA | 0.853 | 0.878 | **0.803** | 0.890 | 0.899 | 0.886 | 1.302 | 1.000 | 1.106 | 0.958 | 0.981 | 1.012 | 1.558 | 1.027 | 1.252 | 15.049 |
| Monthly | SMAPE | **12.549** | 12.556 | 12.641 | 12.670 | 12.791 | 12.677 | 14.588 | 14.014 | 13.514 | 14.260 | 13.917 | 13.958 | 17.642 | 14.062 | 20.128 | 143.237 |
| | MASE | **0.914** | 0.917 | 0.930 | 0.933 | 0.969 | 0.937 | 1.368 | 1.053 | 1.037 | 1.102 | 1.097 | 1.103 | 1.913 | 1.141 | 2.614 | 16.551 |
| | OWA | **0.864** | 0.866 | 0.876 | 0.878 | 0.899 | 0.880 | 1.149 | 0.981 | 0.956 | 1.012 | 0.998 | 1.002 | 1.511 | 1.024 | 1.927 | 12.747 |
| Others | SMAPE | **4.685** | 4.715 | 4.946 | 4.891 | 5.061 | 4.925 | 7.267 | 15.880 | 6.709 | 4.954 | 6.302 | 5.485 | 24.786 | 24.460 | 32.491 | 186.282 |
| | MASE | **3.007** | 3.107 | 2.985 | 3.302 | 3.216 | 3.391 | 5.240 | 11.434 | 4.953 | 3.264 | 4.064 | 3.865 | 18.581 | 20.960 | 33.355 | 119.294 |
| | OWA | **0.983** | 0.986 | 1.044 | 1.035 | 1.040 | 1.053 | 1.591 | 3.474 | 1.487 | 1.036 | 1.304 | 1.187 | 5.538 | 5.013 | 8.679 | 38.411 |
| Weighted Average | SMAPE | **11.618** | 11.698 | 11.807 | 11.829 | 11.927 | 11.851 | 14.718 | 13.525 | 13.639 | 12.840 | 12.780 | 12.909 | 16.987 | 14.086 | 18.200 | 160.031 |
| | MASE | **1.528** | 1.556 | 1.590 | 1.585 | 1.613 | 1.599 | 2.408 | 2.111 | 2.095 | 1.701 | 1.756 | 1.771 | 3.265 | 2.718 | 4.223 | 25.788 |
| | OWA | **0.827** | 0.838 | 0.851 | 0.851 | 0.861 | 0.855 | 1.172 | 1.051 | 1.051 | 0.918 | 0.930 | 0.939 | 1.480 | 1.230 | 1.775 | 12.642 |

for all inputs $\mathbf{x}_{v,t}$ and $i, j, k, \ell \in \{0, 1\}$. Now, starting from $\begin{pmatrix} \mathbf{S}_{0,0}^{(1)} \\ \mathbf{S}_{0,0}^{(2)} \end{pmatrix} = \begin{pmatrix} I & I & 0 \\ I & I & 0 \end{pmatrix}$, we have:

$$\begin{pmatrix} \mathbf{S}_{0,1} \\ \mathbf{S}_{1,0} \end{pmatrix} = \begin{pmatrix} I & I & 0 \\ I & I & 0 \end{pmatrix} \circledast \begin{pmatrix} \mathbf{A}_1 & \mathbf{A}_2 & \mathbf{B}_1 \mathbf{x}_{0,1} \\ \mathbf{A}_3 & \mathbf{A}_4 & \mathbf{B}_2 \mathbf{x}_{1,0} \end{pmatrix} \tag{65}$$

$$= \begin{pmatrix} \mathbf{A}_1 & \mathbf{A}_2 & \mathbf{B}_1 \mathbf{x}_{0,1} \\ \mathbf{A}_3 & \mathbf{A}_4 & \mathbf{B}_2 \mathbf{x}_{1,0} \end{pmatrix}. \tag{66}$$

Re-using operator $\circledast$, we have:

$$\begin{pmatrix} \mathbf{S}_{1,1} \\ \mathbf{S}_{1,1} \end{pmatrix} = \begin{pmatrix} \mathbf{S}_{0,1} \\ \mathbf{S}_{1,0} \end{pmatrix} \circledast \underbrace{\begin{pmatrix} \mathbf{A}_1 & \mathbf{A}_2 & \mathbf{B}_1 \mathbf{x}_{1,1} \\ \mathbf{A}_3 & \mathbf{A}_4 & \mathbf{B}_2 \mathbf{x}_{1,1} \end{pmatrix}}_{\text{Pre-computed}} \tag{67}$$

$$= \begin{pmatrix} \mathbf{A}_1^2 & \mathbf{A}_2^2 & \mathbf{A}_1 \mathbf{B}_1 \mathbf{x}_{0,1} + \mathbf{A}_2 \mathbf{B}_2 \mathbf{x}_{1,0} + \mathbf{B}_1 \mathbf{x}_{1,1} \\ \mathbf{A}_3 & \mathbf{A}_4^2 & \mathbf{A}_3 \mathbf{B}_1 \mathbf{x}_{0,1} + \mathbf{A}_4 \mathbf{B}_2 \mathbf{x}_{1,0} + \mathbf{B}_2 \mathbf{x}_{1,1} \end{pmatrix} \tag{68}$$

Fixing the variate by putting 0 as the initial points of the operator, and then looking at the third element of each row, these elements are calculating the hidden states of the recurrent (it can be shown by a straightforward induction). Accordingly, using this operation, we can recursively calculate the outputs of 2D SSM.

However, using Theorem 3.2, we know that this is an associative operation, so instead of calculating in the recurrent form, we can use parallel pre-fix sum make this computation parallel, decreasing the sequential operations required to calculate the hidden states. Note that since our above operation can model the problem as an parallel prefix, all the algorithms for this problem can be used to enhance the efficiency.

## G.3 Proof of Theorem 3.4

To prove this theorem, we need to (1) show that Chimera can recover SpaceTime. Given this, since SpaceTime is capable of recovering ARIMA [14], exponential smoothing [16], and controllable linear time–invariant systems [87], we can conclude that Chimera can also recover these methods. Then, (2) we need to prove that Chimera can recover SARIMA. This is the model that SpaceTime is not capable of recovering due to the additional seasonal terms.

Note that using $\mathbf{A}_2 = \mathbf{A}_3 = \mathbf{A}_4 = 0$, results in a 1D SSM, with companion matrix as the structure of $\mathbf{A}_1$, which is SpaceTime. Accordingly, SpaceTime is a special case of Chimera when the recurrence only happen along the time direction.

Note that as discussed in Proposition 3.1, multiplying the discretization parameter $\Delta$ results in multiplying the steps. Accordingly, using $s$ as the $\Delta$ in our seasonal module and also letting $\mathbf{A}_2 = \mathbf{A}_3 = \mathbf{A}_4 = 0$ for the seasonal module, we can model the seasonal terms in the formulation of $\text{SAR}(p, q, s)$, meaning that Chimera can also recover SARIMA which is ARIMA with seasonal terms. Note that the reason that Chimera is capable of such modeling is that it uses two heads separately for trend and seasonal terms. Therefore, using different discretization parameters, each can model their own corresponding terms in $\text{SAR}(p, q, s)$.

### G.4  Proof of Theorem 3.5

Similar to the above, using $\mathbf{A}_2 = \mathbf{A}_3 = 0$, our formulation is equivalent to S4D, while we use diagonal matrices as the structure of $\mathbf{A}_1$. Similarly, as discussed by Behrouz et al. [31], MambaMixer is equivalent to S4ND but on patched data. Using our Theorem 5, we can recover linear layers, resulting in recovering TSMixer by setting $\mathbf{A}_2 = \mathbf{A}_3 = 0$.

### G.5  Proof of Theorem 3.6

We in fact will show that restricting Chimera results in recovering 2DSSM [49]. As discussed earlier, this method do not use discretization and initially starts from a discrete system. Also, it uses input-independent parameters. Therefore, we use $\texttt{Linear}_{\Delta_1}(.) = \texttt{Linear}_{\Delta_2}(.)$ as broadcast function, and restrict Chimera to have input-independent parameters, then Chimera can recover 2DSSM [49].

## H  Experimental Settings

We provide the description of datasets in Table 7.

### H.1  Baselines

In our experiments, we use the following baselines:

- Table 8: TSM2 [31], Simba [57], TCN [9], iTransformer [27], RLinear [58], PatchTST [29], Crossformer [26], TiDE [55], TimesNet [8], DLinear [59], SCINet [56], FEDformer [24], Stationary [86], Autoformer [21].
- Table 9: ModernTCN [9], PatchTST [29], TimesNet [8], N-HiTS [65], N-BEATS* [66], ETSformer [20], LightTS [67], DLinear [11], FEDformer [24], Stationary [68], Autoformer [21], Pyraformer [23], Informer [6], Reformer [25], LSTM [69].
- Table 10: LSTM [69], LSTNet [88], LSSL [34], Trans.former [17], Reformer [25], Informer [6], Pyraformer [23], Autoformer [21], Station. [68], FEDformer [24], ETSformer [20], Flowformer [22], DLinear [11], LightTS. [67], TimesNet [8], PatchTST [29], MTCN [9].

For the results of the baselines, we re-use the results reported by Wu et al. [8], or from the original cited papers.

## I  Additional Experimental Results

### I.1  Long Term Forecasting Full Results

The complete results of long term forecasting are reported in Table 8.

### I.2  Short-Term Forecasting

The complete results of short term forecasting are reported in Table 9.

### I.3  Classification

The complete results of time series classification are reported in Table 10.

Table 10: Full results for the classification task (accuracy %). We omit "former" from the names of Transformer-based methods. For all methods, the standard deviation is less than 0.1%.

| Datasets / Models | LSTM [1997] | LSTNet [2018] | LSSL [2022] | Trans. [2017] | Re. [2020] | In. [2021] | Pyra. [2021] | Auto. [2021] | Station. [2022] | FED. [2022] | /ETS. [2022] | /Flow. [2022] | DLinear [2023] | LightTS. [2022] | TimesNet [2023] | PatchTST [2023] | MTCN [2024] | **Chimera** (ours) |
|---|---|---|---|---|---|---|---|---|---|---|---|---|---|---|---|---|---|---|
| EthanolConcentration | 32.3 | 39.9 | 31.1 | 32.7 | 31.9 | 31.6 | 30.8 | 31.6 | 32.7 | 31.2 | 28.1 | 33.8 | 32.6 | 29.7 | 35.7 | 32.8 | 36.3 | 39.8 |
| FaceDetection | 57.7 | 65.7 | 66.7 | 67.3 | 68.6 | 67.0 | 65.7 | 68.4 | 68.0 | 66.0 | 66.3 | 67.6 | 68.0 | 67.5 | 68.6 | 68.3 | 70.8 | 70.4 |
| Handwriting | 15.2 | 25.8 | 24.6 | 32.0 | 27.4 | 32.8 | 29.4 | 36.7 | 31.6 | 28.0 | 32.5 | 33.8 | 27.0 | 26.1 | 32.1 | 29.6 | 30.6 | 32.9 |
| Heartbeat | 72.2 | 77.1 | 72.7 | 76.1 | 77.1 | 80.5 | 75.6 | 74.6 | 73.7 | 73.7 | 71.2 | 77.6 | 75.1 | 75.1 | 78.0 | 74.9 | 77.2 | 81.3 |
| JapaneseVowels | 79.7 | 98.1 | 98.4 | 98.7 | 97.8 | 98.9 | 98.4 | 96.2 | 99.2 | 98.4 | 95.9 | 98.9 | 96.2 | 96.2 | 98.4 | 97.5 | 98.8 | 99.1 |
| PEMS-SF | 39.9 | 86.7 | 86.1 | 82.1 | 82.7 | 81.5 | 83.2 | 82.7 | 87.3 | 80.9 | 86.0 | 83.8 | 75.1 | 88.4 | 89.6 | 89.3 | 89.1 | 89.5 |
| SelfRegulationSCP1 | 68.9 | 84.0 | 90.8 | 92.2 | 90.4 | 90.1 | 88.1 | 84.0 | 89.4 | 88.7 | 89.6 | 92.5 | 87.3 | 89.8 | 91.8 | 90.7 | 93.4 | 93.7 |
| SelfRegulationSCP2 | 46.6 | 52.8 | 52.2 | 53.9 | 56.7 | 53.3 | 53.3 | 50.6 | 57.2 | 54.4 | 55.0 | 56.1 | 50.5 | 51.1 | 57.2 | 57.8 | 60.3 | 59.9 |
| SpokenArabicDigits | 31.9 | 100.0 | 100.0 | 98.4 | 97.0 | 100.0 | 99.6 | 100.0 | 100.0 | 100.0 | 100.0 | 98.8 | 81.4 | 100.0 | 99.0 | 98.3 | 98.7 | 100.0 |
| UWaveGestureLibrary | 41.2 | 87.8 | 85.9 | 85.6 | 85.6 | 85.6 | 83.4 | 85.9 | 87.5 | 85.3 | 85.0 | 86.6 | 82.1 | 80.3 | 85.3 | 85.8 | 86.7 | 86.7 |
| Average Accuracy | 48.6 | 71.8 | 70.9 | 71.9 | 71.5 | 72.1 | 70.8 | 71.1 | 72.7 | 70.7 | 71.0 | 73.0 | 67.5 | 70.4 | 73.6 | 72.5 | 74.2 | **75.3** |

Table 11: Full results for the anomaly detection task. The P, R and F1 represent the precision, recall and F1-score (%) respectively. A higher value of P, R and F1 indicates a better performance.

| Datasets | | SMD | | | MSL | | | SMAP | | | SWaT | | | PSM | | | Avg F1 |
|---|---|---|---|---|---|---|---|---|---|---|---|---|---|---|---|---|---|---|
| Metrics | | P | R | F1 | P | R | F1 | P | R | F1 | P | R | F1 | P | R | F1 | (%) |
| LSTM | [1997] | 78.52 | 65.47 | 71.41 | 78.04 | 86.22 | 81.93 | 91.06 | 57.49 | 70.48 | 78.06 | 91.72 | 84.34 | 69.24 | 99.53 | 81.67 | 77.97 |
| Transformer | [2017] | 83.58 | 76.13 | 79.56 | 71.57 | 87.37 | 78.68 | 89.37 | 57.12 | 69.70 | 68.84 | 96.53 | 80.37 | 62.75 | 96.56 | 76.07 | 76.88 |
| LogTrans | [2019] | 83.46 | 70.13 | 76.21 | 73.05 | 87.37 | 79.57 | 89.15 | 57.59 | 69.97 | 68.67 | 97.32 | 80.52 | 63.06 | 98.00 | 76.74 | 76.60 |
| TCN | [2019] | 84.06 | 79.07 | 81.49 | 75.11 | 82.44 | 78.60 | 86.90 | 59.23 | 70.45 | 76.59 | 95.71 | 85.09 | 54.59 | 99.77 | 70.57 | 77.24 |
| Reformer | [2020] | 82.58 | 69.24 | 75.32 | 85.51 | 83.31 | 84.40 | 90.91 | 57.44 | 70.40 | 72.50 | 96.53 | 82.80 | 59.93 | 95.38 | 73.61 | 77.31 |
| Informer | [2021] | 86.60 | 77.23 | 81.65 | 81.77 | 86.48 | 84.06 | 90.11 | 57.13 | 69.92 | 70.29 | 96.75 | 81.43 | 64.27 | 96.33 | 77.10 | 78.83 |
| Anomaly* | [2021] | 88.91 | 82.23 | 85.49 | 79.61 | 87.37 | 83.31 | 91.85 | 58.11 | 71.18 | 72.51 | 97.32 | 83.10 | 68.35 | 94.72 | 79.40 | 80.50 |
| Pyraformer | [2021] | 85.61 | 80.61 | 83.04 | 83.81 | 85.93 | 84.86 | 92.54 | 57.71 | 71.09 | 87.92 | 96.00 | 91.78 | 71.67 | 96.02 | 82.08 | 82.57 |
| Autoformer | [2021] | 88.06 | 82.35 | 85.11 | 77.27 | 80.92 | 79.05 | 90.40 | 58.62 | 71.12 | 89.85 | 95.81 | 92.74 | 99.08 | 88.15 | 93.29 | 84.26 |
| LSSL | [2022] | 78.51 | 65.32 | 71.31 | 77.55 | 88.18 | 82.53 | 89.43 | 53.43 | 66.90 | 79.05 | 93.72 | 85.76 | 66.02 | 92.93 | 77.20 | 76.74 |
| Stationary | [2022] | 88.33 | 81.21 | 84.62 | 68.55 | 89.14 | 77.50 | 89.37 | 59.02 | 71.09 | 68.03 | 96.75 | 79.88 | 97.82 | 96.76 | 97.29 | 82.08 |
| DLinear | [2023] | 83.62 | 71.52 | 77.10 | 84.34 | 85.42 | 84.88 | 92.32 | 55.41 | 69.26 | 80.91 | 95.30 | 87.52 | 98.28 | 89.26 | 93.55 | 82.46 |
| ETSformer | [2022] | 87.44 | 79.23 | 83.13 | 85.13 | 84.93 | 85.03 | 92.25 | 55.75 | 69.50 | 90.02 | 80.36 | 84.91 | 99.31 | 85.28 | 91.76 | 82.87 |
| LightTS | [2022] | 87.10 | 78.42 | 82.53 | 82.40 | 75.78 | 78.95 | 92.58 | 55.27 | 69.21 | 91.98 | 94.72 | 93.33 | 98.37 | 95.97 | 97.15 | 84.23 |
| FEDformer | [2022] | 87.95 | 82.39 | 85.08 | 77.14 | 80.07 | 78.57 | 90.47 | 58.10 | 70.76 | 90.17 | 96.42 | 93.19 | 97.31 | 97.16 | 97.23 | 84.97 |
| TimesNet (I) | [2023] | 87.76 | 82.63 | 85.12 | 82.97 | 85.42 | 84.18 | 91.50 | 57.80 | 70.85 | 88.31 | 96.24 | 92.10 | 98.22 | 92.21 | 95.21 | 85.49 |
| TimesNet (R) | [2023] | 88.66 | 83.14 | 85.81 | 83.92 | 86.42 | 85.15 | 92.52 | 58.29 | 71.52 | 86.76 | 97.32 | 91.74 | 98.19 | 96.76 | 97.47 | 86.34 |
| CrossFormer | [2023] | 83.6 | 76.61 | 79.70 | 84.68 | 83.71 | 84.19 | 92.04 | 55.37 | 69.14 | 88.49 | 93.48 | 90.92 | 97.16 | 89.73 | 93.30 | 83.45 |
| PatchTST | [2023] | 87.42 | 81.65 | 84.44 | 84.07 | 86.23 | 85.14 | 92.43 | 57.51 | 70.91 | 80.70 | 94.93 | 87.24 | 98.87 | 93.99 | 96.37 | 84.82 |
| ModernTCN | [2024] | 87.86 | 83.85 | 85.81 | 83.94 | 85.93 | 84.92 | 93.17 | 57.69 | 71.26 | 91.83 | 95.98 | 93.86 | 98.09 | 96.38 | 97.23 | 86.62 |
| Chimera | (ours) | 87.74 | 83.29 | 85.46 | 84.01 | 86.83 | 85.39 | 93.05 | 58.12 | 71.55 | 92.18 | 95.93 | 94.01 | 97.30 | 96.19 | 96.74 | **86.69** |

## I.4 Anomaly Detection

The complete results of anomaly detection tasks are reported in Table 11.

## J Additional Ablation Study

## K Comparison with Koopman-based Deep Models

We also compare the results of Chimera with Koopman-based deep models [84]. In ECL dataset, Chimera achieves 0.132, 0.141, 0.144 for $(h = 96, 144, 192)$, while Koopa [84] achieves 0.136, 0.149, 0.156. Similarly, in ETTh2, Chimera achieves 0.262, 0.309, 0.320 for $(h = 96, 144, 192)$, while Koopa achieves 0.297, 0.333, 0.356. Similarly, in Exchange, Chimera achieves 0.077, 0.126, 0.159 for $(h = 96, 144, 192)$, while Koopa achieves 0.083, 0.130, 0.184. Since Liu et al. [84] has focused on these datasets and values for the horizon, we use the same setting for Chimera. The results for Koopa are from its original paper.

Table 12: Ablation Study on additional datasets.

| Method | BVFC | Exchange | | Traffic | |
|---|---|---|---|---|---|
| | Acc. % | MSE | MAE | MSE | MAE |
| Chimera | 58.99 | 0.077 | 0.198 | 0.366 | 0.248 |
| Uni.-directional | 57.29 | 0.091 | 0.203 | 0.369 | 0.255 |
| w/o Gating | 56.18 | 0.094 | 0.210 | 0.373 | 0.259 |
| Input-independent | 55.31 | 0.113 | 0.228 | 0.402 | 0.271 |
| w/o seasonal | 58.12 | 0.083 | 0.202 | 0.372 | 0.258 |
| All Diagonal | 57.98 | 0.095 | 0.209 | 0.370 | 0.257 |
| All Companion | 54.01 | 0.108 | 0.226 | 0.375 | 0.261 |

# L    Limitations

The main goal of this study is to enhance the time series modeling with a wide range of impact on society, from improving the healthcare system using developing deep learning models for analysing medical health records to forecasting stock. We, however, emphasis that our work is a proof-of-concept, meaning that it has error modes.

