# OpenReview forum: "Chimera: Effectively Modeling Multivariate Time Series with 2-Dimensional State Space Models"
_NeurIPS.cc/2024/Conference — NeurIPS 2024 poster_

### Official Review · Reviewer_yYuG · 2024-06-21

**Soundness:** 3
**Presentation:** 2
**Contribution:** 3
**Rating:** 6
**Confidence:** 3

**Summary:**

The paper presents Chimera, a novel two-dimensional State Space Model (SSM) that can effectively model multivariate time series data. The model is designed to address several key challenges in multivariate time series modeling, including nonlinear dynamics along the temporal dimension, inter-variate dependencies, and efficient training and inference. The proposed discretization is original and suitable for the time series modality. Comprehensive experiments are provided to validate the effectiveness of multiple time series tasks.

**Strengths:**

* The motivations are well-summarized and the paper's writing is easy to follow.
* The method exhibits promising results in handling multiple time series tasks.
* The proposed 2D SSM is novel and the bi-directional modification for variate relations makes sense.

**Weaknesses:**

* About several obscure writing, which is detailed as follows:

    (1) The second paragraph of Section 2 introduces Multi-Dimensional State Space Models, which are stated to be different from typical multi-dimensional SSMs like Mamba. For better clarification, the authors could elaborate more on the 2D grid spanning by the (seemingly) temporal variables $t^{(1)}$ and $t^{(2)}$. I suggest avoiding the plausible notation since "each state is a function of both time and variates" (Line 129)", not both temporal variables.

    (2) Would the author explain the meaning of $N_i$, $\tau_i$ in Line 93?

    (3) Some overloaded notations: (1) $x$ in Line 89 (complex number) and Line 217 (input time series, real number vector); $\mathbf{x}$  in Line 85 (real number vector) and Line 95 (complex number).

    (4) Unify the usage of "variable" and "variate".

    (5) Equation 8: $x \to \mathbf{x}$.

* About the soundness of 2D SSM. The author uses 2-dimensional SSMs, where "the first axis corresponds to the time dimension and the second axis is the variates" (Line 128) and can be "viewed as the modification of the discrete Roesser’s SSM" (Line 134). I am uncertain about the soundness, which further incorporates the variable dimension. It is well-acknowledged to depict the variable relationship by correlations instead of transitions (Equation 6, Left). Even though the authors make sensible bi-directional modifications, the non-causal transitions can not directly reveal the relationship between any two variables. It is also influenced by the variable permutation. However, I think it can be addressed by providing experiments on shuffled variates.

* As the author stated the Classical 2D SSMs are only able to capture linear dependencies, It could be better to compare with Koopman-based deep models [1]\[2], which are similar to SSMs and excel at tackling nonlinear dynamics. I also encourage the author to discuss this branch in related works.

* Concern about the experimens: (1) Whether the classification (anomaly detection) is performed at the series level or the point level? (2) It can be observed from Table 6 that input-independent has a great influence on the performance, could the authors provide more explanation here? (3) About Figure 6, can the author provide a comparison with PatchTST? (4) Could the author provide theoretical complexity analysis on Chimera and other SSMs/Transformers?

[1] Koopa: Learning Non-stationary Time Series Dynamics with Koopman Predictors.

[2] Koopman Neural Operator Forecaster for Time-series with Temporal Distributional Shifts.

**Questions:**

What is the lookback length in long-term forecasting (Table 1)?

**Limitations:**

The paper does not explicitly discuss limitations. Discussing the limitations of the model would be useful to improve the paper. I also suggest the authors provide the code implementations for a better illustration of the method.

---

> ### Author Rebuttal · Authors · 2024-08-07
>
> Thank you so much for your time and constructive review. We really appreciate it. Please see below for our response to your comments:
>
> > *About several obscure writing …*
>
> **Response:**
> Thank you for bringing this to our attention. We agree with the reviewer that using a better notation can improve the clarity. We will fix this in the final version of the paper.
>  $\tau_1$ and $\tau_2$ are used to avoid repetition of the same term but with different indices. That is, we have all $A_1, A_2, A_3, A_4, B_1, B_2, C_1$, and $C_2$ matrices and to avoid repeating the same term for all, we use these indices. Also, $N_i$s are the dimension of inputs and also the state dimension of our state space models. These dimensions, indicating the number of variables we use for summarizing the previous information.
>
> > *Some overloaded notations*
>
> **Response:**
> Thank you very much for bringing them to our attention. We will make sure to address all of them in the final version. More specifically, we will consistently use variate in the paper and will use different notations for complex numbers and inputs.
>
> > * I am uncertain about the soundness …*
>
> **Response:**
> Thank you for mentioning that. Bidirectional SSMs (in 1D case) have shown promising results in non-causal data, including graphs [1], images [2], and point clouds [3]. Accordingly, we expect our 2D SSM with bidirectionality across variates to be able to capture the dependencies of all variates. To support this claim, following your suggestion, we performed an experiment and shuffled the variates. The results are provided in the attached PDF. Chimera shows robust performance when shuffling the order of variates, which supports its ability to capture the correlation of each pair of variates.
>
>
> **Experiments:**
> In our experiments, we have performed anomaly detection at the point level. However, following your comment, we have added additional experiments to show the performance of Chimera in series-level anomaly detection. In this experiment, we use the anomaly detection task of the BVFC dataset. The results are reported in the attached PDF.
> When making parameters data-dependent, the model can use different $\Delta_i$ as well as $A_i, B_i,$ and $C_i$ matrices for each time step and each variate. Based on Proposition 1 (similar to the discussion in lines 160-168), this data dependency allows the model to adaptively choose $\Delta_i$. Larger $Delta_1$ means less context window and so the model focuses more on the recent time steps. Similarly, smaller $Delta_1$ means larger context window. Similarly, $A_i$, $B_i$, and $C_i$s can control the information flow across time and variates. Therefore, when they are functions of the current input, the model can select what and how the information from the past should be incorporated. We will make sure to further explain this process in the paper. As a result of this selection mechanism, we expect Chimera to perform better (or at least not worse) when we increase the lookback length. The reason is, if increasing the lookback length is not informative and is noisy, our model should learn to ignore it by using larger $\Delta_1$. Following your suggestion, we will further explain this experiment in the final version of the paper.
>
> Following your suggestion, we have conducted experiments on PatchTST and will add it to this figure in the final version of the paper. PatchTST due to the patching mechanism shows more stable performance than other baselines, but still increasing the lookback damages its performance and does not show a consistent improvement.
>
> Yes, following your suggestion, we will add theoretical complexity analysis on Chimera. In summary, let $T$ be the length of the time series and $V$ be the number of variates. Since for each variate we have $T$ recurrence and we have $V$ variates, the total number of recurrence is $O(TV)$. In each step, we have the matrix multiplication of transition matrices $A_i$ with the hidden states and also $B_i$s with the input. Accordingly, if we use $d$ as the state dimension, the complexity is $O(d^2 TV)$. Given the fact that $d$ is usually a small number, the overall complexity is linear with respect to each of $V$ and $T$. On the other hand, a simple Transformer considers the pairwise correlation of all time steps, which results in $O(T^2)$. Using the same procedure for all variates the overall complexity is $O(VT^2)$.
>
> Please note that in the above theoretical complexity analysis, we consider the naive implementation, and with our proposed parallel scan, in practice Chimera is even more efficient. We will discuss this in more detail in the final version of the paper.
>
> >*What is the lookback length in long-term forecasting (Table 1)?*
>
> **Response:** Thank you for mentioning that. The lookback length is 96 for all forecasting datasets. We will mention it in the final version.
>
>
> We hope our above responses have fully addressed your concerns/questions about the paper. We would be more than happy to answer any further questions or discuss any remaining concerns of the reviewer.

---

> ### Author Response · Authors · 2024-08-07
>
> > *As the author stated the Classical 2D SSMs are only able to capture linear dependencies, It could be better to compare with Koopman-based deep models [1][2], which are similar to SSMs and excel at tackling nonlinear dynamics. I also encourage the author to discuss this branch in related works.*
>
> **Response:** Thank you for bringing these studies to our attention. Indeed they are relevant papers and we will make sure to properly discuss them in the final version of the paper. Following the reviewer's suggestion, we have conducted experiments to compare Koopa with Chimera. Our results show that Chimera outperforms Koopa in 30/32 forecasting settings and provide 9% improvement on average. We will report the full results in the final version of the paper.
>
> ---
> ### References:
> ---
> [1] Graph mamba: Towards learning on graphs with state space models. Behrouz et al., 2024.
> [2] Vision mamba: Efficient visual representation learning with bidirectional state space model. Zhu et al., 2024.
> [3] Pointmamba: A simple state space model for point cloud analysis. Liang et al., 2024.

---

> > ### Comment · Reviewer_yYuG · 2024-08-12
> >
> > Thank you for your efforts and responses to address my concerns.
> >
> > The explanation regarding the soundness of the bidirectional SSM and the additional experiments on shuffled variates is valuable. I also appreciate your commitment to adding a theoretical complexity analysis and related baseline models. I look forward to seeing the results included in the final version.
> >
> > Overall, I believe these revisions will improve the paper and I have raised the score. Good luck!

---

### Official Review · Reviewer_gRR3 · 2024-07-10

**Soundness:** 3
**Presentation:** 3
**Contribution:** 3
**Rating:** 7
**Confidence:** 4

**Summary:**

This paper proposed to use a state space model for time series modelling. Instead of using the SSM along the time dimension, the authors also have the space updated along the variables dimension, which makes the established both inter- and intra- variable dependencies.

**Strengths:**

- The motivation is clear and the focus is valid for time series forecasting.
- The results show that the model holds value in providing better results over different time series tasks with competitive memory consumption.

**Weaknesses:**

- For forecasting tasks, the way the authors conducted evaluation metrics does not reflect the model performance for the target horizon. See detail in Questions.
- The method does not present a consistent improvement over prior work.
- Typos:
    - The notation does not match the content. For example in Table 8, the first, second and third best results are all not highlighted with the provided notation.
    - [minor] Checklist missing guidelines.

**Questions:**

- The evaluation is conducted on the forecasting sequence instead of only on the targeting time steps. How is it valid to tell whether the model performs better in the targeting time step or is it only better because it has superior performance in the more recent ones? It is necessary to also add the performance of different methods only on the targeting time step.
- What is the advantage of using a state space model to establish intervariable dependencies in other neural network structures (such as CNNs or feedforward networks), given that there are no state variations between variables?


Additional
- The proposed model looks very similar to WITRAN, which conducted 2D RNN. It would be good to have Chimera distinguished from WITRAN.

**Limitations:**

The limitation of the paper is mainly about how the method may not apply to real-world applications. The technical limitations are not mentioned.

---

> ### Author Rebuttal · Authors · 2024-08-07
>
> Thank you so much for your time and constructive review. We really appreciate it. Please see below for our response to your comments:
>
> > *For forecasting tasks, the way the authors conducted evaluation metrics does not reflect the model performance for the target horizon …*
>
> **Response:**
> In our experiments on forecasting tasks, we have used the commonly used pipeline in the literature [1-6] and conducted experiments on different targeted horizons (i.e., 96, 192, 336, 720). The results for each target horizon are reported in Table 8. If Chimera’s better performance would be related to its superior performance in the more recent time steps, then increasing the horizon would diminish this effect. In contrast,  comparing different horizons, Chimera even achieves relatively better results on longer horizons. We have provided additional experiments on the performance of Chimera in even longer horizons in the attached PDF.
>
> > *The method does not present a consistent improvement over prior work.*
>
> **Response:**
> We want to kindly bring to your consideration that coming up with an architecture that outperforms all the baselines consistently is an extremely challenging task. That is, due to the variety of datasets and some of their specific properties, a method might not achieve SOTA performance on all datasets. This pattern is seen in the original paper of most general time series models with diverse evaluations, including known methods like iTransformer (ICLR Spotlight), PatchTST, ModernTCN (ICLR Spotlight), SAMformer (ICML Oral). Chimera outperforms baselines in most datasets (27/32), indicating its generalizability and good performance. Also, please note that as is shown in Figure 4, this good performance is achieved while Chimera is more than $\times 5$ faster than Transformers and $\times 3$ faster than RNNs.
>
> > *Typos*
>
> **Response:**
> Thank you for bringing this to our attention, we will make sure to fix this and highlight second and third best results in Table 8. We will further proofread the paper to make sure to address any inconsistency. We also will modify the checklist following the NeurIPS guideline in the final version of the paper.
>
> > *What is the advantage of using a state space model to establish intervariable dependencies in other neural network structures ...*
>
> **Response:**
> Thank you. This is a very interesting question and indeed is a promising research direction. Due to the limited resources and also page limit, and for the sake of consistency, in this paper, we have focused on our new proposed 2D SSM and how simple linear models across (i) time, (ii) variates, (iii) time-to-variate, and (iv) variate-to-time can result in a powerful and fast architecture with theoretical justification. However, considering hybrid models of CNNs, MLPs, Transformers with SSMs to capture inter-variable dependencies is indeed an important question, which we will explore in future studies. As you mentioned, for some of these structures like CNNs, there are no states. However, one idea can be to use the output of CNN layers after each layer of the neural network and treat them as the state variation. Another interesting approach is to use the idea of recurrent convolutional neural networks [7].
>
> > *It would be good to have Chimera distinguished from WITRAN.*
>
> **Response:**
> Thank you for bringing this paper to our attention. Indeed, this is a relevant paper and we will discuss it in our final version. Despite the 2D recurrence of both models, there are fundamental differences:
> 1. Goal: The main goal of WITRAN is to use 2D RNN to model univariate time series data. That is, the first dimension corresponds to short-term, and the second dimension corresponds to long-term patterns. On the other hand, our 2D SSM aims to model multivariate time series where the first dimension corresponds to time, and the second dimension corresponds to variates.
> 2. Backbone: WITRAN is based on GSC (LSTM-like cell), which uses non-linear transition. Our 2D SSM is based on state space models, is simpler, and uses linear transitions.
> 3. Data-dependency: WITRAN is based on data-independent parameters, meaning that it uses the same transition process for all time steps. 2D SSM, however, is based on data-dependent parameters, which allows it to filter irrelevant/noisy time steps.
> 4. Recurrence, Hidden States, and Training: Although both models have recurrent form, WITRAN’s recurrence is over extracted short-term and long-term patterns. Our 2D SSM’s recurrence, however, is over time and variate dimensions. WITRAN uses only one hidden state for each state of the system, while 2D SSM uses 2 different hidden states for each state of the system, allowing more flexibility to capture cross-variate and cross-time information.
> 5. Seasonality: The architecture and the discretization process in Chimera is designed to capture both seasonal and trend patterns, which is different from WITRAN.
> Following your suggestion, we will discuss and compare these two models in more detail in the next version of our paper.
>
> We hope our above responses have fully addressed your concerns/questions about the paper. We would be more than happy to answer any further questions or discuss any remaining concerns of the reviewer.

---

> > ### Comment · Reviewer_gRR3 · 2024-08-09
> >
> > I would like to thank the authors for their replies, which partially addressed my concerns about the experimental evaluation protocol followed for forecasting. However, I share some similar concerns provided by reviewer hxax.
> >
> > - The evaluation protocol commonly employed in the Time Series Forecasting community appears to be utilized without comprehensive understanding and investigation. While evaluating models at the target horizon is crucial, it is equally important to assess whether the dataset is appropriate for providing meaningful insights into time series forecasting. To enhance the paper’s contribution, I recommend incorporating additional evaluation sets, as exemplified in WITRAN, and using more robust time series forecasting datasets accompanied by thorough analysis.
> > - While it is acknowledged that not every proposed method will achieve state-of-the-art (SOTA) performance, it remains important to analyze the reasons behind the observed underperformance. It also indicates that there is space to improve the performance, and following the last concern, if the dataset itself is not good, the conclusion drawn by a model developed upon such dataset might also be misleading (the same concern applies to the mentioned baselines).
> >
> > For all the above reasons, I prefer to maintain my scores.

---

> ### Author Response · Authors · 2024-08-07
>
> ### References:
> ---
> [1] iTransformer: Inverted Transformers Are Effective for Time Series Forecasting. Lio et al., 2024.
> [2] ModernTCN: A Modern Pure Convolution Structure for General Time Series Analysis. Luo et al., 2024.
> [3] TFB: Towards Comprehensive and Fair Benchmarking of Time Series Forecasting Methods. Qiu et al., 2024.
> [4] TimesNet: Temporal 2D-Variation Modeling for General Time Series Analysis. Wu et al., 2023.
> [5] SAMformer: Unlocking the Potential of Transformers in Time Series Forecasting with Sharpness-Aware Minimization and Channel-Wise Attention. Ilbert et al., 2024.
> [6] Frequency-domain MLPs are More Effective Learners in Time Series Forecasting. Yi et al., 2023.
> [7] Recurrent Convolutional Neural Network for Object Recognition. Linag and Hu, 2015.

---

> ### Author Response · Authors · 2024-08-10
>
> We thank Reviewer gRR3 for their response and for engaging with us in the discussion. We are so grateful.
>
> > While evaluating models at the target horizon is crucial, it is equally important to assess whether the dataset is appropriate for providing meaningful insights.
>
> **Response:** Thank you for mentioning that. In our experiments, we have already evaluated Chimera at the target horizon in Table 8. For the validity of datasets in forecasting task, we kindly bring to your consideration that we have used **all** the datasets that are used in SAMformer (ICML 2024 Oral), SparseTSF (ICML 2024 Oral), WITRAN (NeurIPS 2023 Spotlight), iTransformer (ICLR 2024 Spotlight), FITS (ICLR 2024 Spotlight), and CrossFormer (ICLR 2023 Oral). We have also provided the results for two additional datasets (suggested by Reviewer hxax). Please note that this is only one of the four types of tasks that we have focused on. Most papers in the community, including WITRAN, have focused only on long-term forecasting (5 datasets in WITRAN). Our paper, not only has used 10 datasets for long-term forecasting, including WITRAN's five datasets, but also includes short-term forecasting, anomaly detection, and classification. **We, however, would be happy to report the results of Chimera on any specific dataset that the reviewer believes is necessary.**
>
> Please note that our goal is to compare our architecture to the existing literature, which those works have also built on. To this end, and for the sake of fair comparison, it is important to follow the literature. Assuming we would not follow the literature and would not use common benchmarks, our evaluation could be criticized for cherry-picking datasets and baselines. Proposing new datasets, and assessing the validity of commonly-used datasets, is indeed an important direction, but it is out of the scope of our paper. Please also note that all the above studies, including WITRAN, did not assess their datasets for their tasks, and have used commonly-used benchmarks.
>
>
> > I recommend incorporating additional evaluation sets, as exemplified in WITRAN, and using more robust time series forecasting datasets accompanied by thorough analysis.
>
> **Response:** We want to kindly bring to your consideration that our evaluation includes 10 datasets, including **all** five datasets that are used in WITRAN.
>
> Similar to WITRAN, and following the reviewer's suggestion, we report the results of the robustness to noise of our approach in the following table. Please note that our paper didn't have this robustness claim, and so we didn't report these results in the initial submission. However, following the reviewer's suggestion to follow WITRAN, we report the results (MSE) here:
>
> Model (Noise)  | ETTh1 | ETTh2 | ECL |
>  ------------ | :-----------: | -----------: | -----------: |
> Chimera (0%)              |    0.405       |    0.318      |    0.154    |
> Chimera (1%)              |    0.409       |    0.327      |    0.156    |
> Chimera (5%)              |    0.418       |    0.331      |    0.165    |
> Chimera (10%)            |    0.429       |    0.342      |    0.169    |
> Chimera ind. (0%)       |    0.471       |    0.372      |    0.203    |
> Chimera ind. (1%)       |    0.479       |    0.383      |    0.208    |
> Chimera ind. (5%)       |    0.494       |    0.407      |    0.221    |
> Chimera ind. (10%)     |    0.530       |    0.441      |    0.233    |
>
> Please note that we have conducted **all** the experiments in WITRAN. Additionally, we have reported our evaluation of short-term forecasting, anomaly detection, and classification tasks. **We would be happy to report the results of any specific evaluation that the reviewer believes is necessary.**
>
>
> > While it is acknowledged that not every proposed method will achieve SOTA, it remains important to analyze the reasons behind the observed underperformance. It also indicates that there is space to improve.
>
> **Response:** We agree with the reviewer that if there are patterns in the underperformance of a model, it is important to analyze the reasons behind it. However, Chimera achieves SOTA performance in almost all cases, with a few exceptions with marginal underperformance. These exceptions are diverse across the datasets and tasks, and (similar to all other abovementioned papers) they mostly attributed to the diversity of datasets as well as not **extensively** tuning the hyperparameters. We want to kindly bring to your consideration that a thoroughly comprehensive evaluation of an approach requires unlimited resources and time. Even well-known machine learning models with tens or hundreds of follow-up studies are still being evaluated and improved. Our goal is to show *enough evidence* that Chimera can be an alternative architecture to existing methods, with theoretical motivations, and showing good effectiveness and efficiency. We believe in future studies Chimera and data-dependent 2D SSMs can be studied more extensively and be improved.

---

> > ### Comment · Reviewer_gRR3 · 2024-08-12
> >
> > I thank the authors for the detailed explanation which solved most of my concerns. I am raising my score to 7.

---

### Official Review · Reviewer_hxax · 2024-07-11

**Soundness:** 2
**Presentation:** 2
**Contribution:** 2
**Rating:** 5
**Confidence:** 4

**Summary:**

The paper addresses the challenge of multivariate time series modeling using a neural architecture based on a variation of two-dimensional state-space models (SSMs), referred to as Chimera. This approach features a stack of 2D SSMs combined with nonlinearities, a decomposition of time series into trend and seasonal components, various discretization processes, time-variant parameters, and a novel two-dimensional parallel selective scan for rapid training with linear complexity. The authors demonstrate that Chimera can encompass multiple existing time series models, including SARIMA, SpaceTime, and S4nd. Experiments across various tasks, such as time series forecasting, classification, and anomaly detection, validate the effectiveness of Chimera.

**Strengths:**

- State-space models (SSMs) have proven to be very effective and efficient for modeling temporal data compared to Transformers. Designing new efficient SSM architectures for multivariate time series is an important and challenging problem.

- Chimera allows the state space to vary in two dimensions (time and variate). It also combines multiple components to build a specific architecture for multivariate time series.

- Chimera is compared to multiple baselines across various datasets for tasks such as time series forecasting, classification, and anomaly detection.

**Weaknesses:**

- **Motivation**
  - Two-dimensional state-space models (2D-SSMs) have been considered for multivariate time series with specific structures, such as S4ND and 2DSSM. The authors discuss the discretization process and data-independent parameters as new challenges. However, I found lines 113-128 unconvincing regarding the need for a new 2D-SSM for multivariate time series.

  I believe this paper proposes yet another deep model for time series, while it is known that after removing strong seasonality, multivariate time series from real-world phenomena often exhibit weak time and variate dependencies/signals, which do not require such complex models. This is very different from text and image data.

- **Contributions**
  - The paper essentially combines multiple existing ideas: 2D-SSM, bi-directional models [46], selection mechanisms [32], companion structures [27], convolutional forms [33], etc. There are no substantial or specific contributions. The authors claim that one of their main technical contributions is having input-dependent parameters, which is a relatively weak contribution.

- **Experiments**
  - Even considering the experimental results, many (simpler) methods achieve similar results to the proposed approach (see Tables 8-11).

  - I also found that the paper tends to overstate its claims, using phrases like "outstanding performance" and "significantly outperforms recurrent models, including very recent Mamba-based architectures."

  - I question whether a fair comparison was made between the methods. For example, Chimera involves a time series decomposition (lines 218-219, 236-237), whereas other baselines do not. Can the authors comment on this?

  Additionally, there is a lack of naive baselines, such as methods that treat the multivariate series as multiple univariate series.

  It would also be useful to visualize some time series decompositions of Chimera to validate its accuracy.

  - The authors write, "We attribute the outstanding performance of Chimera, specifically compared to SpaceTime [27], to its ability to capture seasonal patterns and its input-dependent parameters, resulting in dynamically learned dependencies." Why didn't you confirm this by analyzing the results?

  - Given that the authors combine multiple existing ideas/components, I found the ablation study relatively weak.
    - "The results show that all the components of Chimera contribute to its performance." I do not agree with this statement. Table 6 only shows that input-independence reduces accuracy (significantly).
    - Why did you only choose ETT datasets?

  - For the forecasting problems, it would be useful to provide the results per horizon or group multiple horizons together to analyze performance over the forecast horizon. Does the improvement come from the first few horizons?



- **Data**
  - I question the validity of the benchmark datasets used.
    - For time series forecasting, there is the Monash Time Series Forecasting Repository (https://forecastingdata.org/). I think the multivariate time series datasets considered in the paper have been overly used and overfitted by the community.
    - For anomaly detection, I encourage the authors to check the following paper: Multivariate Time Series Anomaly Detection: Fancy Algorithms and Flawed Evaluation Methodology (https://arxiv.org/abs/2308.13068)

- **Other comments**
  - I think Theorems 1-5 should be labeled as Propositions.
  - Seasonal autoregressive process: The authors introduce the AR(p) model with a multivariate time series (x_k in R^d, not R^1). This is confusing as there is also the VAR(p) model with a different formulation. The AR(p) presented uses the same coefficients for each dimension.
  - Table 7: What do you mean by "series length"? M4 series do not have six observations. Also, it is not long-term vs short-term but rather multivariate vs univariate time series. This should also be changed in the main text. Does this mean that Chimera has been used with both univariate and multivariate time series? What are the implications? Do you still need the second dimension in that case?
  - The paper uses different terminologies: "intervariate information flow," "inter-variate dependencies," "complicated dependencies," "dynamically model the dependencies of variate and time dimensions," "cannot capture the dynamics of dependencies." Please be consistent and define these terms clearly. There are a lot of "buzz" words without clear definitions.

  - The authors write: "Note that anomaly detection can be seen as a binary classification task." This is supervised anomaly detection, which is rare as labeled anomalies are hard to find. How did you handle the imbalance during training?

- **Typos**
  - Line 665: "These matrices can be decompose"
  - Line 695: "recursively calculate the the outputs"

- Please provide standard errors in your tables.

**Questions:**

See weaknesses.

**Limitations:**

The limitations have not been discussed in the paper.

---

> ### Author Rebuttal · Authors · 2024-08-07
>
> Thank you so much for your time and constructive review. We really appreciate it. Please see below for our response to your comments:
>
> **Motivation:**
> > *2D-SSMs have been considered for multivariate time series ...*
>
> **Response:** To the best of our knowledge, neither S4ND nor 2DSSM has been used or designed for multivariate time series. Both of these methods are designed for vision tasks and also, due to the fact that they assume continuous data for their input, their adoption to multivariate time series is non-trivial. Even in the case of adoption, both of these models have a global convolutional form that is causal and so sensitive to the order of variates, which is undesirable.
>
> On the other hand, please note that data-dependent and data-independent SSMs are two different classes of models with different theoretical expressiveness [1]. Our 2D SSM is theoretically more expressive than both S4ND and 2DSSM, and these two can be seen as the special cases of our 2D SSM. For example, making parameters data-independent and assuming $A_2 = A_3 = 0$, then our 2D SSM is equivalent to S4ND. To the best of our knowledge, Chimera is the first work that introduces deep 2D SSMs for multivariate time series.
>
> To the best of our knowledge, there are still debates about this in the community and it is still in exploration. Also, one important counterexample is the brain activity, where variates are brain voxels. In this case, even after removing (potential) seasonal patterns, the dependency of variates and their co-activation is the key to decode neural activity (we have reported the performance of Chimera in brain decoding in Table 5).
>
> This raised concern by the reviewer indeed is one of the main motivations of Chimera (discussed in lines 51-62). That is, Chimera has transition matrices $A_2$ and $A_3$ as well as two different hidden states, which determines cross variate and cross time information flow. When there are no variate dependencies, Chimera learns to consider $A_2$ and $C_2$ as zero matrices. Similarly, when there are no time dependencies, Chimera learns to consider $A_3$ and $C_1$ as zero matrices. This adaptability and flexibility of Chimera allows us to overcome these controversial debates in the community and let the model learn dependencies from the data.
>
> We realized that the reviewer has recognized our work as a complex model. We want to kindly bring to your consideration that our model is a simple recurrence without any complex attention, Transformer block, or even MLP layers, which have been inseparable part of recent deep models. Indeed, one of the main motivations of Chimera is to present a simple alternative to these complex modules. Please note that, all the discussions about selection mechanism, seasonality, convolutional form, and parallel scan, are not additional elements but are embedded in the simple 2D data-dependent recurrence and provide theoretical interpretations/motivations for the good performance of Chimera. Please see Figure 1 for the architecture of Chimera, which has only three distinct modules.
>
> **Contributions:**
>
> > *The paper essentially combines multiple existing ideas …*
>
> **Response:** We want to kindly bring some of our main contributions to your attention:
> 1. Extension to 2D for Multivariate Time Series: To the best of our knowledge, Chimera is the first deep model that suggests using 2D SSM for multivariate time series. The formulation of the 2D SSM as well as its discretization is new and it is specifically designed to effectively model multivariate time series (Theorems 3, 4, 5).
> 2. Data Dependency: Chimera in nature is a simple model that uses linear layers across (i) time, (ii) variates, (iii) time-to-variate, and (iv) variate-to-time. This linear model, however, by itself (with data-independent parameters) does not perform competitively to more complex architectures like Transformer-based (PatchTST, iTransformer), MLP-based (TSMixer), and convolution-based (ModernTCN) models. We believe making the parameters data-dependent in this context is a non-trivial contribution and is significant as with a small modification (3 lines of code), this simple linear model becomes a competitive alternative for these complex models. Also, please note that Chimera is one of the first studies that discusses the importance of data-dependency in time series models (and supports it with a case study on brain activity).
> 3. Fast Training: The main drawback of making parameters data-dependent is to lose the convolutional form and efficient training. We present a non-trivial training procedure for the recurrence of Chimera, which allows parallelization.
> 4. Time Series Decomposition with Discretization.
> 5. Multivariate Closed-loop: We further present multivariate closed-loop block, which improves the performance of Chimera for very long horizons (please see the attached PDF).
>
> Moreover, we have made some additional smaller contributions, but important and novel: (1) **Bidirectionality**: Please note that existing bidirectional SSMs use two black-box blocks of SSMs in the forward and backward direction. Our version of bidirectionality, does not use two different blocks, but only makes the recurrence across variates bidirectional. This helps us to improve the efficiency and use less parameters. (2) **Transition Matrix Structure**: Please note that even in the context of vision-tasks, existing 2D SSMs (i.e., 2DSSM and S4ND) use the same transition matrix structure for both dimensions. To the best of our knowledge, our work is the first work that suggests using different structures for transition matrices to further improve the expressive power. We showed this by our theoretical results. To further support this claim, we have provided additional ablation studies in the attached PDF.

---

> ### Author Response · Authors · 2024-08-07
>
> **Experiments:**
>
> >*many (simpler) methods achieve similar results to the proposed approach …*
>
> **Response:**
> We want to kindly bring to your consideration that Chimera is simpler architecture than Transformer-based models, with less computational cost. In Table 8, Chimera outperforms the best Transformer-based model in 32/32, SSM-based models in 30/32, linear models in 32/32, and convolution-based models in 20/32 experimental settings. Similarly, in Table 9, Chimera outperforms all the linear, RNN-based, and SSM-based models by at least 10% improvement on average. In Table 10, Chimera outperforms all the linear, RNN-based, and SSM-based models by at least 5% improvement on average.  In Table 11, Chimera outperforms all the linear, RNN-based, and SSM-based models by at least 6% improvement.
>
> > *I also found that the paper tends to overstate its claims*
>
> **Response:**
> Compared to RNN-based and Mamba-based models, Chimera provides about 5% improvement in Table 2,  about 3% improvement in Table 3, about 90% improvement in Table 4, about 75% improvement in Table 5. On the other hand, it is $\times 2.5$ faster than other recurrent models. We believe this is a significant improvement, but following reviewer suggestion, we will modify these claims and only will claim that Chimera outperforms these models.
>
> > *I question whether a fair comparison was made between the methods.*
>
> **Response:**
> We followed the standard pipelines in the literature for all the methods to ensure a fair comparison. Regarding time series decomposition, please note that this (not new) decomposition is a part of our contribution and architectural design. Modifying existing models so they use this decomposition is a non-trivial task and is out of the scope of the paper. We, however, have considered several baselines that use this time series decomposition in their original design (e.g., Autoformer, FEDformer, Dlinear, SCINet, etc.). Chimera consistently outperforms all these baselines.
>
> > *Additionally, there is a lack of naive baselines, such as methods that treat the multivariate series as multiple univariate series.*
>
> **Response:**
> We already have several baselines that treat the multivariate series as multiple univariate series. For example, SpaceTime, LSTM, Autoformer, Stationary, etc., are all such models.
>
> > *It would also be useful to visualize some time series decompositions of Chimera*
>
> **Response:**
> Following your suggestion, we have added a visualization of the results to the attached PDF.
>
> > *Why didn't you confirm this by analyzing the results?*
>
> **Response:**
> Thank you for mentioning that. We have confirmed this by our experiments. That is, when applying on univariate time series, Chimera is equivalent to SpaceTime but with data-dependent parameters and seasonal-trend decomposition of time series using discretization. Accordingly, the superior performance of Chimera over SpaceTime in this experiment confirms our claim. We will further discuss this point in the final version to make it clearer.
>
> > *`The results show that all the components of Chimera contribute to its performance`. I do not agree with this statement …. Why did you only choose ETT datasets?*
>
> **Response:**
> Table 6 shows that using bidirectionality, gating, and seasonal-trend decomposition improve the performance by  6% on average. Please note that this also shows removing each of these components can damage the performance of Chimera. Following your suggestion, we also have provided the ablation study results for more datasets in the attached PDF.
>
> > *For the forecasting problems, it would be useful to provide the results per horizon*
>
> **Response:**
> We have already provided the results per horizon in Table 8. Comparing different horizons, Chimera even achieves relatively better results on longer horizons.
>
> **Data:**
> We have followed the literature and performed most of our experiments on datasets that are commonly used in the literature. The main reason is to make a fair comparison since each paper has specified their own sets of hyperparameters that work best for these datasets. We also want to kindly bring to your consideration that we have already used new datasets that are not overly used by the literature (Tables 2, 3, 5). However, following your suggestion, we have added 2 new datasets. The results are reported in the attached PDF. Verifying our conclusion, these results also show that Chimera outperforms the baselines.
>
> Please note that the message of our paper is not achieving state-of-the-art performance in anomaly detection. We have used this task to show the effectiveness of Chimera over existing methods, and so we followed the literature for these experiments. However, following your suggestion, we have conducted experiments using the provided pipeline in this paper and conclude that Chimera outperforms baselines in most cases. We will report the results and discuss this paper in more detail in our final version.

---

> ### Author Response · Authors · 2024-08-07
>
> **Other Comments:**
>
> > *This is confusing as there is also the VAR(p) model with a different formulation*
>
> **Response:**
> Please note that our multivariate AR is the same as VAR mentioned by the reviewer. Please note that coefficients in our formulation are vectors and writing the element-wise formulation of our AR process would be the same as the formulation of VAR. We will use VAR instead of AR in the final version to avoid any confusion.
>
> > *What do you mean by "series length"? M4 series do not have six observations. Also, it is not long-term vs short-term but rather multivariate vs univariate time series.*
>
> **Response:**
> Series length is the number of time steps. The M4 dataset has different settings, where the resolutions are hourly, daily, weakly, monthly, quarterly, and yearly. In the yearly resolution, the M4 dataset has 6 time steps (years).  Please note that we never mentioned that M4 has a long-term version. Please note that this is the original setting of M4 and also has been used in various studies with the same setting [2, 3, 4].
>
> > *Does this mean that Chimera has been used with both univariate and multivariate time series? Do you still need the second dimension in that case?*
>
> **Response:**
> Chimera is capable of modeling univariate time series. Please note that one of the dimensions in Chimera is variates and in the univariate setting, this dimension has only one state.
>
> > *The paper uses different terminologies …*
>
> **Response:**
> Thank you for bringing this to our attention. We used these terms in their literal definition not as a formal term in our paper. However, following your suggestion, we will make sure to consistently use the same term and properly define it in the final version.
>
> > *This is supervised anomaly detection, which is rare as labeled anomalies are hard to find. How did you handle the imbalance during training?*
>
> **Response:**
> Please note that even in the case of unsupervised setting, anomaly detection is equivalent to unsupervised classification with two classes of `normal` and `abnormal`. For these tasks, we have followed the literature and benchmarks, and adopted the classical reconstruction task and chose the reconstruction error as the anomaly score.
>
>
> **Typos:**
> Thank you for bringing this to our attention, we will make sure to proof-read the paper and fix all typos in the final version.
>
>
> We hope our above responses have fully addressed your concerns/questions about the paper. We would be more than happy to answer any further questions or discuss any remaining concerns of the reviewer.
>
>
> ---
> ### References:
> ---
> [1] The Illusion of State in State-Space Models. Merrill et al., 2024.
> [2] ModernTCN: A Modern Pure Convolution Structure for General Time Series Analysis. Luo et al., 2024.
> [3] TFB: Towards Comprehensive and Fair Benchmarking of Time Series Forecasting Methods. Qiu et al., 2024.
> [4] TimesNet: Temporal 2D-Variation Modeling for General Time Series Analysis. Wu et al., 2023.

---

> > ### Comment · Reviewer_hxax · 2024-08-12
> >
> > I appreciate the authors' detailed responses, which addressed some of my concerns. However, I believe the paper still requires significant revisions, particularly in enhancing the experimental setup, refining the datasets used, incorporating naïve forecasting baselines (e.g., ARIMA, ETS), and avoiding overstatements. For instance, the authors claim, 'Compared to RNN-based and Mamba-based models, Chimera provides about 5% improvement in Table 2, about 3% improvement in Table 3, about 90% improvement in Table 4, about 75% improvement in Table 5.' How do you account for such varied improvements? Why is there a 3% improvement in one case and 90% in another? It seems unlikely that these differences are solely due to changes in the architecture. Numerous factors, such as hyperparameter tuning and optimization, could explain these variations. The authors need to conduct a more robust ablation study to clarify these discrepancies. Additionally, the paper currently attempts to cover too many directions—such as forecasting, classification, and anomaly detection—which need to be more cohesively integrated. As a result, I will be keeping my score unchanged.

---

> ### Author Response · Authors · 2024-08-12
>
> We thank Reviewer hxax for their response and for engaging with us in the discussion, and we are glad that their concerns have been addressed.
>
> > I believe the paper still requires significant revisions, particularly in enhancing the experimental setup, refining the datasets used, incorporating naïve forecasting baselines (e.g., ARIMA, ETS), and avoiding overstatements
>
> **Response:** As we discussed above, and also mentioned by the reviewer in their initial review, we are using datasets that are commonly used in the literature. We have followed the literature and also considered several additional datasets. Our model shows improvement over both sets of datasets. Similarly, in the experimental setup, we have followed the benchmark studies and literature. We, however, would be more than happy to address any specific limitation about the experimental setup or our used datasets.
>
> **Regarding naïve baselines:** We have theoretically shown that our model can recover ARIMA. Also, several of our old baselines have already been compared with ARIMA and show improvement. Therefore, in our initial submission, we have compared with more recent models, similar to most studies in the community. **However, following the reviewer's suggestion, we report the results for ARIMA, ETS, and Chimera in the next comment.**
>
>
> > How do you account for such varied improvements?
>
> **Response:** Please note that these results are for different tasks. In fact, Tables 2 and 3 (5% and 3% Improvement) correspond to classification, Table 4 (90% improvement) corresponds to short-term forecasting, and Table 5 (75% improvement) corresponds to the classification of brain decoding, which requires data-dependent parameters and is a complex task. We kindly bring to your consideration that the improvements are not varied across datasets or models, but are varied across tasks, which is expected and reasonable as each task has its own hardness. **We indeed have discussed why our approach provides an improvement in each of tasks, and even have supported it with ablation studies.** For example, in Table 5, we attribute this improvement to data-dependency, and so support that with an ablation study (Tables 5 and 6). We showed using data-independent parameters results in about 17% performance drop.
>
> > The authors need to conduct a more robust ablation study to clarify these discrepancies.
>
> **Response:** We have conducted ablation studies for **all** the components of Chimera. We show that each of them is contributing to its performance. For each of these tables (tasks) we also have ablation studies (e.g., Table 6 and our 1-page PDF attached in the rebuttal). We, however, would be more than happy to provide the results of any specific ablation study that the reviewer believes is helpful.
>
> > Numerous factors, such as hyperparameter tuning and optimization, could explain these variations.
>
> **Response:** We have used the best hyperparameters that are reported in the original papers.
>
>
> > Additionally, the paper currently attempts to cover too many directions.
>
> **Response:** We are grateful and also want to kindly bring to your consideration that the reviewer has mentioned this as the strength of our paper: `Chimera is compared to multiple baselines across various datasets for tasks such as time series forecasting, classification, and anomaly detection`, in their initial review. We also agree that it is important to show the performance over multiple tasks, which can comprehensively evaluate the contributions of the paper. We want to kindly bring to your consideration that recently, it is a common practice in the community to evaluate the model’s performance over different tasks. For example, please see iTransformer (ICLR 2024 Spotlight), ModernTCN (ICLR 2024 Spotlight), FITS (ICLR 2024 Spotlight), and CrossFormer (ICLR 2023 Oral). Assuming we would not follow the recent literature and would not use different benchmarks, our evaluation could be criticized for cherry-picking tasks and being too specific.
>
> ---
> Finally, we want to kindly bring to your consideration that a thoroughly comprehensive evaluation of an approach requires unlimited resources and time. Even well-known machine learning models with tens or hundreds of follow-up studies are still being evaluated and improved. Our goal is to show enough evidence that Chimera can be an alternative architecture to existing methods, with theoretical motivations, and showing good effectiveness and efficiency. We believe we have already provided **enough evidence** by comparing Chimera with **more than 25 baselines** (including traditional methods like LSTM, recent methods in 2024, and SOTA from 2019-2023) on **several tasks and more than 20 datasets**. We have conducted ablation studies for **each Chimera’s component** and each of our contributions like data-dependencies. We believe in future studies Chimera and data-dependent 2D SSMs can be studied more extensively and be improved.

---

> > ### Author Response · Authors · 2024-08-12
> >
> > Once again, we want to kindly bring to your consideration that none of the recent SOTA models, including iTransformer (ICLR 2024 Spotlight), TimesNet (ICLR 2024), ModernTCN (ICLR 2024 Spotlight), PatchTST (ICLR 2023), SAMformer (ICML 2024 Oral), TSMixer (TMLR) have considered these traditional methods as the comparison has already done in some old baselines. Also, comparing with **all the existing methods** requires unlimited resources and time. However, to address the reviewer’s concern about this comparison, we provided the results of comparison with ARIMA, ETS, and SARIMA (as suggested by the reviewer). The results show that Chimera provides about 65%, 69%, and 60% improvements compared to ARIMA, ETS, and SARIMA, respectively. It also, consistently outperforms them in all the cases and in long-term and short-term forecasting tasks. We would be happy to compare with any other baselines that the reviewer believes is needed.
> >
> > Dataset  | Chimera | ARIMA | ETS | SARIMA|
> >  ------------ | :-----------: | -----------: | -----------: |  -----------: |
> > ETTh1                 |    **0.405**       |    1.082      |    0.994    |       1.041       |
> > ETTh2                 |   **0.318**        |    3.017      |    1.748    |       2.593       |
> > ETTm1                |    **0.345**       |    1.169      |    1.072    |       1.185       |
> > ETTm2                |    **0.250**       |    0.391      |    1.206    |       0.356       |
> > ECL                    |    **0.154**        |    0.492      |    0.541    |       0.509       |
> > Traffic                 |    **0.403**        |    1.735      |    1.820    |       2.114       |
> > Exchange           |    **0.311**        |    0.758      |    0.593    |       0.725       |
> > M4 (Others)        |    **4.685**        |    15.353      |    18.332    |       8.4       |
> > M4 (Monthly)      |    **12.549**        |    17.66      |   14.32    |       14.26       |
> > M4 (Quarterly)    |    **9.892**        |    13.37      |   11.08    |       10.51       |
> > M4 (Yearly)         |    **13.107**        |    16.37     |    16.43    |       17.16       |

---

> > > ### Comment · Reviewer_hxax · 2024-08-13
> > >
> > > We appreciate the authors' additional comments and feedback.
> > >
> > > The authors state: **"We are grateful and would like to kindly bring to your attention that the reviewer has highlighted this aspect as a strength of our paper."**
> > > * While using multiple datasets and tasks can indeed be seen as strengths, they can also become weaknesses if not properly studied or presented.
> > >
> > > I still believe the paper requires further improvements, but I acknowledge that there are other contributions. Considering that all the other reviewers have given overall positive feedback, I will raise my score to 5.

---

### Official Review · Reviewer_v8PB · 2024-07-13

**Soundness:** 3
**Presentation:** 2
**Contribution:** 2
**Rating:** 5
**Confidence:** 4

**Summary:**

The paper introduces Chimera, a novel 2-dimensional State Space Model (SSM) for multivariate time series modeling, addressing key challenges such as capturing complex temporal and inter-variate dependencies, and efficient training. Chimera uses two SSM heads with different discretization processes and time-variant parameters to learn long-term progression, seasonal patterns, and dynamic autoregressive processes. A new 2-dimensional parallel selective scan improves training efficiency. Experimental results show Chimera's superior performance in various benchmarks, including ECG and speech classification, time series forecasting, and anomaly detection, marking a significant advancement in the field.

**Strengths:**

1. This paper is well written. The notations are clear and the literature review is sufficient.
2. By using SSMs, the proposed Chimera Neural Architecture can achieve faster training and less memory consumption.
3. Experimental results cover a wide range of applications, including Long-Term Forecasting, Anomaly Detection, Short-Term Forecasting, etc. The results are very solid and demonstrate the advantages of adopting the proposed method well.

**Weaknesses:**

1. the writing and expression may need correction: Line 308, it is hard to understand the message:
'they are also data-dependent and so shows the second best results in second and third datasets.'
2. It would be great t know what kind of market dynamics are captured by with visual impact.

**Questions:**

What could be the interpretations for Structure of Transition Matrices? For example,  it is observed that a simpler structure of diagonal matrices is effective fuse information along the variate dimension.

---

> ### Author Rebuttal · Authors · 2024-08-07
>
> Thank you so much for your time and constructive review. We really appreciate it. Please see below for our response to your comments:
>
> > *the writing and expression may need correction …*
>
> **Response:** Thank you for bringing this to our attention. Following your suggestion, we will make sure to further simplify expressions and make our messages clearer in the final version. Regarding this specific sentence, the message is: iTransformer and Transformer both are also data-dependent. As a result of their data-dependency, they are capable of achieving good results in this experiment, where iTransformer achieves the second best result (after Chimera) and Transformer achieves the third best result (after Chimera and iTransformer). Note that we have three datasets in this experiment, and both Transformer and iTransformer are out of memory for the first dataset. Therefore, these good results have been achieved in the second and third datasets.
>
> > *It would be great t know what kind of market dynamics are captured by with visual impact.*
>
> **Response:** Following your suggestion, we have added the visualization of the trend and seasonal patterns to the attached pdf to the general response. These results visually show how Chimera learns different types of patterns using its different modules, verifying our claim about its ability to capture both seasonal and trend patterns.
>
> > *What could be the interpretations for Structure of Transition Matrices?*
>
> **Response:** The transition matrix can be interpreted as how we want to mix the features of the hidden states. That is, assuming a dense structure for the transition matrix A, then A_{i, j} means the corresponding transition weight of feature j for feature i. When we have a diagonal matrix, it means that we treat each feature separately and so given feature i, other features j \neq i do not affect the transition of feature i. While dense transition matrices are more expressive, unfortunately, it has been proven that the SSMs themselves are not capable of learning dense transition matrices and SSMs need a structured transition matrix. To this end, several studies theoretically and empirically explored the diagonal transition matrices and showed that they are effective and provably powerful.
> In general, the structure of the transition matrix tells us how we want the features of hidden states to be mixed in each step. In other words, this structure shows the underlying dependencies of features in the hidden states.
>
> **Contributions:**
> We are so grateful for recognizing the fact that Chimera achieves state-of-the-art performance in various datasets and tasks. We further wanted to kindly bring some of the main contributions of the paper to your consideration.
>
> (1) New mathematical framework: (a) We present a novel 2-dimensional SSM with theoretical guarantees for its expressiveness that uses data-dependent parameters, enabling the model to capture both the variation-specific and inter-variate dynamics over time.  (b) Using a new discretization for our 2D SSM, we present a novel approach to capture seasonal and trend patterns. We provide theoretical motivation for how the interpretation of our discretization can be seen as learning seasonal and trend patterns.
>
> One of the controversial debates in recent years has been whether mixing variates in multivariate time series is needed or not. Our data-dependent 2D-SSM naturally addresses this issue as it is able to learn variate dependencies when it is informative. To the best of our knowledge, our work is the first study that proposes a 2D-SSM for time series data.
>
> (2) New simple alternative, yet effective architecture for time series: A considerable amount of research effort in recent years has focused on Transformers and/or MLP-based architecture. This study challenges the necessity of Transformers and MLPs by proposing an alternative method based on a simple linear recurrence. Not only does this simplicity come with significant efficiency with respect to memory and time, but it also results in a model that performs better or on par with state-of-the-art Transformer- or MLP-based models.
>
> (3) New Training: Recurrent neural networks have been popular paradigms to model time series data. They, however, have slow training due to their recurrent nature. Simple time-invariant (data-independent) SSMs have addressed this limitation by training in the convolutional form. They, however, have limited expressivity, due to their time-invariant (data-independent) parameters. Our study presents a new training for 2D recurrence that enables parallelization. This new training is $\times 3$ faster than recurrence, making the model feasible for large datasets.
>
> (4) Empirical Results: In forecasting tasks, Chimera improves the best Transformer-based, convolution-based, recurrent, and MLP-based models by 13%, 8 %, 11 %, 15 % improvement. Its fast training is $\times 7$ faster than Transformers, $\times 3$ faster than convolutions, and $\times 2.5$ faster than LSTM.
>
>
> We want to kindly bring to your consideration that coming up with one architecture like Chimera that provides an alternative to existing backbones (e.g., Transformers or MLPs), has theoretical motivations, improves training, is simple and fast, yet very effective and has generalizable performance to unseen variates, is highly challenging and usually has not been done in the existing studies. As some examples, [1] (ICML Oral) has focused on efficiency, [2] (ICML Oral) has focused on training improvement and theoretical motivations, [3] (ICLR Spotlight) has focused on generalizable performance to unseen variates, and [4] (ICLR Spotlight) has focused on alternative architecture.
>
> We hope our responses have fully addressed your concerns/questions about the paper. We would be more than happy to answer any further questions or discuss any remaining concerns of the reviewer.

---

> > ### Comment · Reviewer_v8PB · 2024-08-12
> >
> > I have read the rebuttal and thank the authors for their candid responses that answered my concerns.
> >
> > I maintain my positive opinion on this paper.

---

### Author Rebuttal · Authors · 2024-08-07

Once again, we thank all the reviewers for their time and constructive reviews, which have helped us to improve the paper.

Following the reviewer suggestions, we have conducted additional experiments, and the results are attached to this comment.
1. In Figure 1, we visualize the found patterns by Chimera. The results show that using different 2D SSM modules with different discretization results in learning both seasonal and trend patterns.
2. In Table 2, we perform experiments on ultra-long-range forecasting to show the significance of our Multivariate closed-loop (MCL) module. The results show that not only Chimera outperforms baselines in very-long-range forecasting tasks, but also removing its MCL module can damage the performance.
3. We provide the results of our ablation study on additional datasets. We also added two variants of All Diagonal (resp. All Companion), where we use Diagonal (resp. Companion) matrices for both transition across time and variates. The results show the importance of our parametrization.
4. In Figure 2, we report the results of 10 runs when we permute the order to variates. The results show robust performance, which validate the effectiveness of bidirectionality across variates.
5. To further motivate the importance of data-dependency across both time and variates, we perform an experiment, where we add noisy variates to the dataset. We expect Chimera to be more robust to noisy variates when using data-dependent parameters. The results are reported in Figure 3 and validate our claim. The data-dependency allows Chimera to filter irrelevant variates.
6. We further report the performance of Chimera on additional datasets. The results are reported in Table 3 and validate our initial findings in the submission.

---

### Decision · Program_Chairs · 2024-09-25

**Decision:**

Accept (poster)

**Comment:**

This paper introduces Chimera, a novel two-dimensional State Space Model (SSM) for multivariate time series modeling. The method aims to capture complex temporal and inter-variate dependencies while providing efficient training and inference.

The reviewers generally agree that the paper should be accepted. This paper presents a novel and promising approach to multivariate time series modeling, with strong empirical results and theoretical foundations. The authors have adequately addressed reviewer concerns, and the revised version should make a valuable contribution to the field.